# Diffraction-engineered holography: Beyond the depth representation limit of holographic displays

Daeho Yang[1], Wontaek Seo[1], Hyeonseung Yu [1], Sun Il Kim[1], Bongsu Shin[1], Chang-Kun Lee[1], Seokil Moon[1], Jungkwuen An [1], Jong-Young Hong[1], Geeyoung Sung[1] & Hong-Seok Lee [1,2] ✉

Holography is one of the most prominent approaches to realize true-to-life reconstructions of objects. However, owing to the limited resolution of spatial light modulators compared to static holograms, reconstructed objects exhibit various coherent properties, such as content-dependent defocus blur and interference-induced noise. The coherent properties severely distort depth perception, the core of holographic displays to realize 3D scenes beyond 2D displays. Here, we propose a hologram that imitates defocus blur of incoherent light by engineering diffracted pattern of coherent light with adopting multi-plane holography, thereby offering real world-like defocus blur and photorealistic reconstruction. The proposed hologram is synthesized by optimizing a wave field to reconstruct numerous varifocal images after propagating the corresponding focal distances where the varifocal images are rendered using a physically-based renderer. Moreover, to reduce the computational costs associated with rendering and optimizing, we also demonstrate a network-based synthetic method that requires only an RGB-D image.

Holography is a recording and reconstruction process based on the interference of multiple wave fields[1]. Holograms duplicate the wave field of the recorded object under an appropriate illumination and provide true-to-life reconstructions of three-dimensional (3D) objects[2]. Beyond the reproduction of a recorded object, the computer-generated hologram (CGH), which is a numerically calculated hologram of a wave field of non-existing objects, enables the display of arbitrary 3D scenes and provides monocular depth cues, unlike traditional displays[3].

Although holographic displays are free from vergence-accommodation conflict, which causes visual fatigue[4] and a significant reduction in the depth constancy[5], unsolved issues originating from their limited resolution still remain. A real-world object scatters light by reflecting light in various directions from the substructures of its rough surface[6], and a static hologram can represent such substructures with a large effective number of pixels[7]. In contrast, dynamic

holograms, of which the resolution is 3 orders of magnitude smaller than that of static holograms[8], cannot spread light without the noise because the interference between voxels becomes noticeable as the number of voxels increases[9,10]. From this perspective, dynamic holograms can be categorized into two different types, namely diffusive holograms and non-diffusive holograms (Fig. 1a).

Diffusive holograms spread light up to the maximum diffraction angle bounded by a pixel pitch by introducing high-frequency patterns[10-14]. For example, high-frequency patterns can be included in holograms by placing voxels with sufficient separation between them[10,11], applying random phases[12,13], and employing point-based methods with physically correct phases[14]. In diffusive holograms, 3D objects can be seen at any position within a viewing angle and out-of-focus objects are blurred as real-world objects. However, the image quality is limited by a small number of points or interference between the points, displaying speckles on the reconstructed scenes[9,10].

[1]Samsung Advanced Institute of Technology, Samsung Electronics, Suwon, Gyeonggi-do, South Korea. [2]Department of Electrical and Computer Engineering, Seoul National University, Seoul, South Korea. ✉e-mail: lhs12100@snu.ac.kr

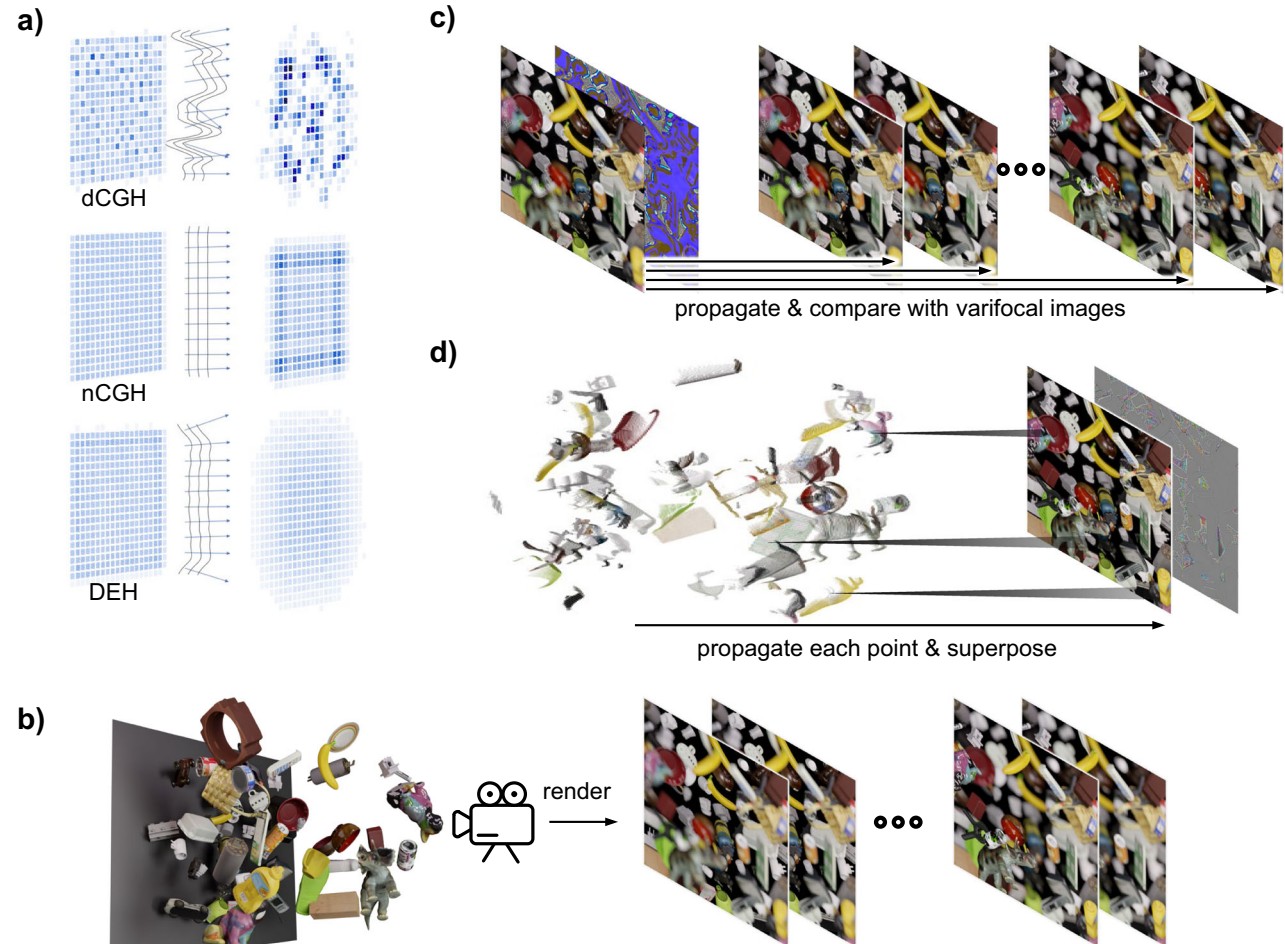

**Fig. 1 | Schematics of diffraction-engineered holography. a** The intensity distributions of the diffusive hologram (dCGH), the non-diffusive hologram (nCGH), and the DEH are drawn for two different planes when a blue square is reconstructed on the left side. The black lines represent the phases of the various holograms, while the blue arrows represent the propagating direction of the light. In the DEH, the content-dependent phase at the edge of the square spreads light over a wide angle so defocus blur can be formed at the other focal plane. **b** Upon varying the focal distance of the camera, varifocal images are rendered using a physically based renderer. **c** The wave field is optimized to satisfy all varifocal images at each depth in the DEH. **d** The nCGH synthesizes a hologram by propagating each pixel at a different depth and superposing the propagated points.

In contrast to diffusive holograms, non-diffusive holograms concentrate on enhancing the image quality of reconstructed scenes. In this case, the position-dependent phase offset is imposed in point-based methods to avoid the rapid phase variation of different depth objects[15,16], phase-retrieval algorithms are adopted to reconstruct single-depth images[17,18], and quadratic phases are utilized to suppress the speckles[19,20]. Although non-diffusive holograms tend to exhibit an enhanced image quality, the coherent properties of light become conspicuous due to a reduced numerical aperture and content-dependent defocus pattern[19,21]. For instance, constructive and destructive interference patterns appear in the intensity distribution of non-diffusive holograms according to Fresnel propagation and those interference patterns are far different from the defocus blur of a real-world object. The inconsistent defocus patterns destroy the relationship between the depth and the blur, which is crucial in the context of depth perception[22,23]. Moreover, the presence of a lucid boundary at the interface between objects with different depths due to interference distorts the perception of the relative depth between objects[24]. Thus, for the high image quality hologram without distortion of depth perception, both advantages of diffusive holograms and non-diffusive holograms are required.

On the other hand, multi-plane hologram attracts great attention recently, especially for their improvement in image quality and computation time[25–29]. For instance, non-convex optimization is adopted to minimize a custom cost function[25], dynamic adjustment of amplitude-constraint is employed to improve image quality[26], and a new algorithm based on singular value decomposition of the Fresnel impulse response function is proposed to enhance computational speed[27]. Since the multi-plane hologram is synthesized by optimizing the wave field to reconstruct one image at one focal plane while optimizing to reconstruct other images at other focal planes, multi-plane holograms are widely adopted in dynamic 3D projections[30–32]. However, experimental realization of high image quality reconstruction with a single wave field is still challenging[30–33].

Here, we demonstrate a diffraction-engineered hologram (DEH) that presents photorealistic scenes and real-world-like defocus blur, enhancing depth expressions of holographic displays by utilizing multi-plane holograms. We take advantage of the fact that the phase variation of light does not affect the image seen by the eyes, but steers the propagating direction of light. Contrary to most of the conventional CGH algorithms, which only optimize the intensity at object-existing planes[15–20], DEH also optimizes diffracted patterns at out-of-focus planes by adaptively changing the phase to enhance defocus blur while leaving the intensity at the object-existing planes nearly the same. To find the phase satisfying such diffracted pattern, the approach of multi-plane hologram[25–31,33,34], which reconstructs

different images depending on a propagation distance, is adopted. The wave field of our hologram is optimized to reconstruct sharply focused images of an object at the object plane and reconstruct blurred images at other focal planes. To obtain blurred and focused images employed as optimization targets, varifocal images are rendered by a physically based renderer that properly handles occluded objects and provides an accurate blur circle similar to that of a human eye. As a result, the DEH achieves both superiorities, namely the image quality of non-diffusive holograms and the depth expression of diffusive holograms. Furthermore, to reduce the computational cost associated with the rendering of varifocal images and the optimization of a complex wave field, we design and train a convolutional neural network. The diffraction-engineered hologram network (DEHNet) synthesizes the complex wave field displaying appropriate blurred images depending on the focal distances while requiring only an RGB-D image as an input. Finally, we confirm the properties of the DEH through simulations and experiments to demonstrate an enhanced depth expression compared to conventional CGHs.

## Results

### Loss function for hologram synthesis

Assuming that a wave field at the $z = 0$ planes is given by $|A(x,y)|e^{i\phi(x,y)}$, the propagated wave field at the $z = d_n$ plane calculated by the angular spectrum method (ASM)[35] is given as

$$\mathrm{Prop}_{d_n}(|A(x,y)|e^{i\phi(x,y)}) = F^{-1}\left\{F\left\{|A(x',y')|e^{i\phi(x',y')}\right\}e^{ik_z d_n}\right\}, \quad (1)$$

where $F(F^{-1})$ is the Fourier (inverse Fourier) transform operator, $e^{ik_z d_n}$ is a propagation kernel with $k_z = \sqrt{k^2 - k_x^2 - k_y^2}$, and $k_x(k_y)$ is the angular wavenumber along the $x(y)$ direction. Here, a notable point of Eq. (1) is the fact that the propagation kernel $e^{ik_z d_n}$ does not alter the amplitude distribution in the Fourier domain, and so the amplitude distribution in the Fourier domain is sustained for every propagation distance. Considering that the diffraction angle is proportional to the spatial frequency[19,21], the application of a wide frequency range of phases is the only means to achieve sufficient defocus blur unless the intensity itself is composed of a wide range of frequencies.

However, the majority of high-quality non-diffusive CGH (nCGH) algorithms fix the phase as zero or as a position-dependent formula[15,16] to avoid speckles, thereby leaving the content-dependent defocus pattern unsolved. The DEH starts from this point. DEH is calculated by optimizing a wave field to possess a content-dependent phase so that the propagated wave field forms a clear image at the object-existing plane while forming a blurred image at other planes. As a target image for each propagated distance, we used varifocal images generated by a rendering process by changing the focal distance of a camera to ensure that blur considering occluded surfaces is efficiently reflected (Fig. 1b). After simulating the propagated intensity of the wave field using the ASM, we calculated the mean square error (MSE) between the propagated intensity and the varifocal image of which the focal distance is equal to the propagation distance (Fig. 1c). The wave field is compared with tens of varifocal images and it is updated using a gradient descent method. The optimization is iterated until the change of the wave field is negligible.

Compared to other researches[16,36] employing learning-based methods or optimization methods, occluded surfaces and defocus blur can be reflected on the reconstructed scene by means of explicitly comparing the propagated intensities and defocused images. Furthermore, to reconstruct sharply focused objects, the wave field is also compared with an all-in-focus image when the propagation distance is close to the depth of the objects (see Section 9 of the Supplementary

material for further details of all-in-focus loss). Standard phase retrieval algorithms, e.g. the iterative Fourier transform algorithm, can be used in multi-plane holograms[30,33,34], but gradient descent optimization is employed to compare the wave field with the depth-weighted all-in-focus image.

In summary, the total loss function $\mathcal{L}$ for optimization is given by

$$\mathcal{L} = \sum_{n=1}^{N}\left[\left\langle\left||\mathrm{Prop}_{d_n}(|A(x,y)|e^{i\phi(x,y)})|^2 - I_{d_n}\right|^2\right\rangle \right.$$
$$\left. + \beta\left\langle\left|\left(|\mathrm{Prop}_{d_n}(|A(x,y)|e^{i\phi(x,y)})|^2 - I_{\mathrm{AIF}}\right)e^{-\left(\gamma\frac{D_n-d_n}{d_0-d_N}\right)^2}\right|^2\right\rangle\right], \quad (2)$$

where $N$ is the number of varifocal images, $I_{d_n}$ is the intensity of the varifocal image at a focal distance $d_n$, $I_{\mathrm{AIF}}$ is the intensity of the all-in-focus image, $D_n$ is the depth map normalized from $d_0$ to $d_N$ with a focal distance $d_n$, $\beta$ is the user-defined loss weight, and $\gamma$ is the user-defined depth attention weight. Here, the depth map with defocus blur depending on the focal distance is used instead of an all-in-focus depth map to reflect the occluded surfaces (see the "Methods" section and Section 10 of the Supplementary material for further details). The first term in Eq. (2) represents the MSE of the propagated wave field compared to the varifocal images, while the second term represents the MSE of the propagated wave field compared to the depth-weighted all-in-focus image. In contrast to a DEH, conventional methods[16] construct holograms by propagating each 3D point for a particular distance depending on its depth value and superposing the propagated points (Fig. 1d). The method simulates the propagation of the points by the ASM and also handles occluded surfaces by ignoring the backside wavefront when the backside and frontside wavefronts overlap.

Holograms depicting a scene with different-sized cubes were synthesized and in-focus (out-of-focus) conditions of the holograms were simulated as shown in Fig. 2a. Even in the out-of-focus conditions, the defocus blur of the nCGH cannot be seen, especially for the large cube, due to the content-dependent defocus pattern[19,21]. Coherent propagation of the wave field forms a Fresnel diffraction pattern which differs from the defocus blur of incoherent light so the depth perception can be distorted[22]. In contrast, the out-of-focus image of the DEH displays a clear defocus blur even if the diameter of the blur circle is slightly smaller than that of the rendered image. However, the most significant drawback of the DEH is its computational load, since a number of varifocal images are required in addition to an optimization procedure. Since nCGH can be synthesized using only RGB-D images, DEHs are not practical in the majority of real-time applications.

To overcome such issues, a neural network (DEHNet) is trained to obtain a DEH from RGB-D images (Fig. 2b). The network is composed of 34 convolutions with 12 channels except for the last layer which includes a concatenated shortcut. Non-linearity and a wide receptive field are more important than hidden features so the number of channels is selected as small as possible to increase the number of convolutions and activations under a restricted computation resource (see Section 8 of the Supplementary material for image quality dependency on the number of channels). The training dataset consists of 3000 different scenes and each of these scenes contains 21 varifocal images, an all-in-focus color image, 21 varifocal depth maps, and an all-in-focus depth map (see Section 6 of the Supplementary material for the minimum number of planes required in DEHNet and Section 11 of the Supplementary material for results with a different number of planes). After training, the weights of the network were quantized to 8-bit integers to reduce computational load. The DEHNet can synthesize an optimal wave field that can reconstruct appropriately blurred and sharply focused images while considering occluded surfaces, and

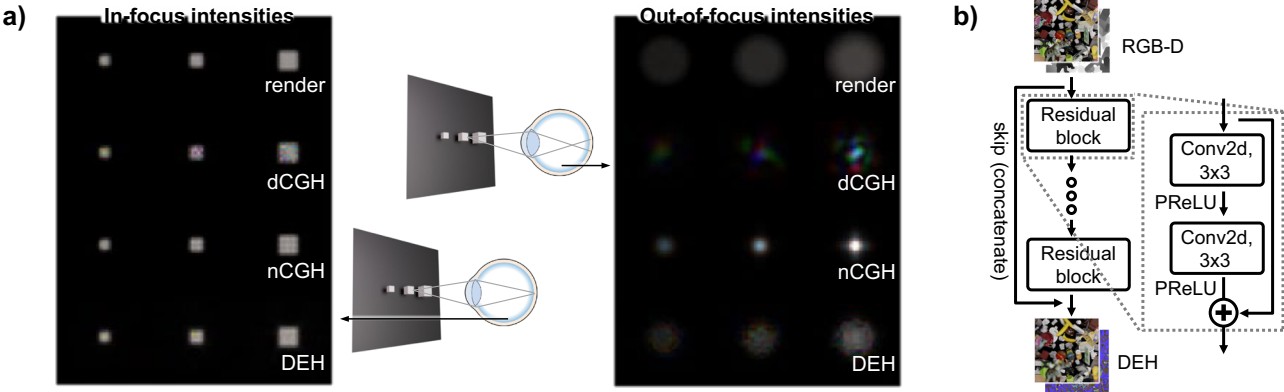

**Fig. 2 | Reconstructed intensities of different algorithms and schematics of DEHNet. a** In-focus and out-of-focus intensities of the rendered case, the dCGH, the nCGH, and the DEH are simulated (from top to bottom). The side lengths of the cubes are 3, 4, and 6 pixels (from left to right). **b** The convolutional neural network synthesizes a wave field from an all-in-focus image and an all-in-focus depth map.

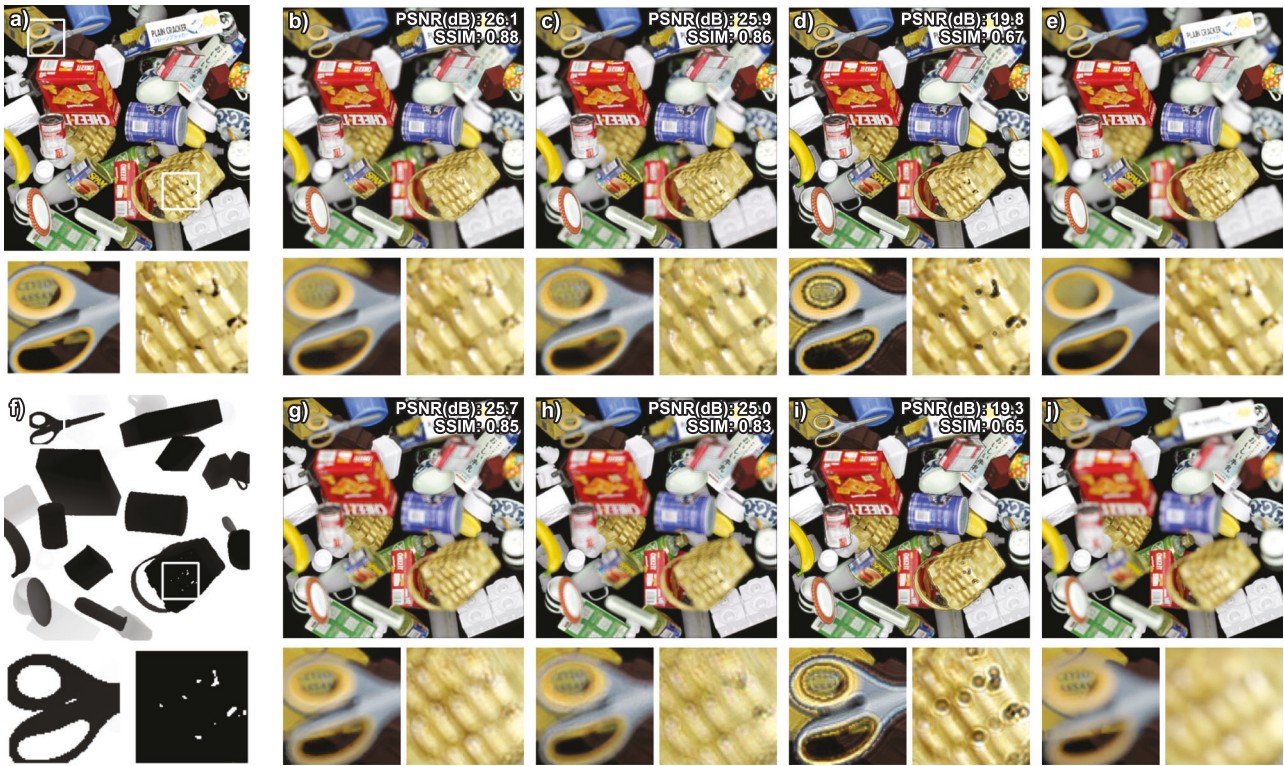

**Fig. 3 | Simulation results for the DEH, the DEHNet, and the nCGH.** All-in-focus rendered image (**a**), depth map (**f**), front focus rendered image (**e**), and rear focus rendered image (**j**). Reconstructed images of DEH (**b, g**), DEHNet (**c, h**), and nCGH (**d, i**). With the exception of the all-in-focus image and the depth map, the top images (**b–d**) correspond to the front focus images and the bottom images (**g–j**) correspond to the rear focus images. The PSNR values (in dB) and the SSIM values are marked on the top right corner of each image. The smaller images represent enlarged views of the larger images. The ASM was used to simulate different focal planes.

this can be achieved using only an all-in-focus color image and an all-in-focus depth map.

## Image quality of reconstructed holograms

Figure 3 shows the simulated results for the DEH, the DEHNet, and the nCGH when the focus is adjusted to the frontside or backside of the scene. We only compared non-diffusive holograms in this paper because the purpose of the paper is synthesizing high-image-quality holograms. A comparison between diffusive holograms and DEH can be found in Section 2 of the Supplementary material. One of the differences between the nCGH and the DEH is the vivid boundary at the interface of the objects which are located at different depths as shown in the enlarged image in Fig. 3. An abrupt phase variation at the interface leads to two coherent beams with different phases coinciding at the interface; the constructive and destructive interferences then build a sharp boundary. Since blurred and sharply focused edges at the occluded surface boundary are used to judge the relative depths between objects[24], the presence of a distorted blur at an edge can be considered one of the most serious defects. Moreover, when a hole exists in an object, the hole is distorted by the depth difference between the object and the background.

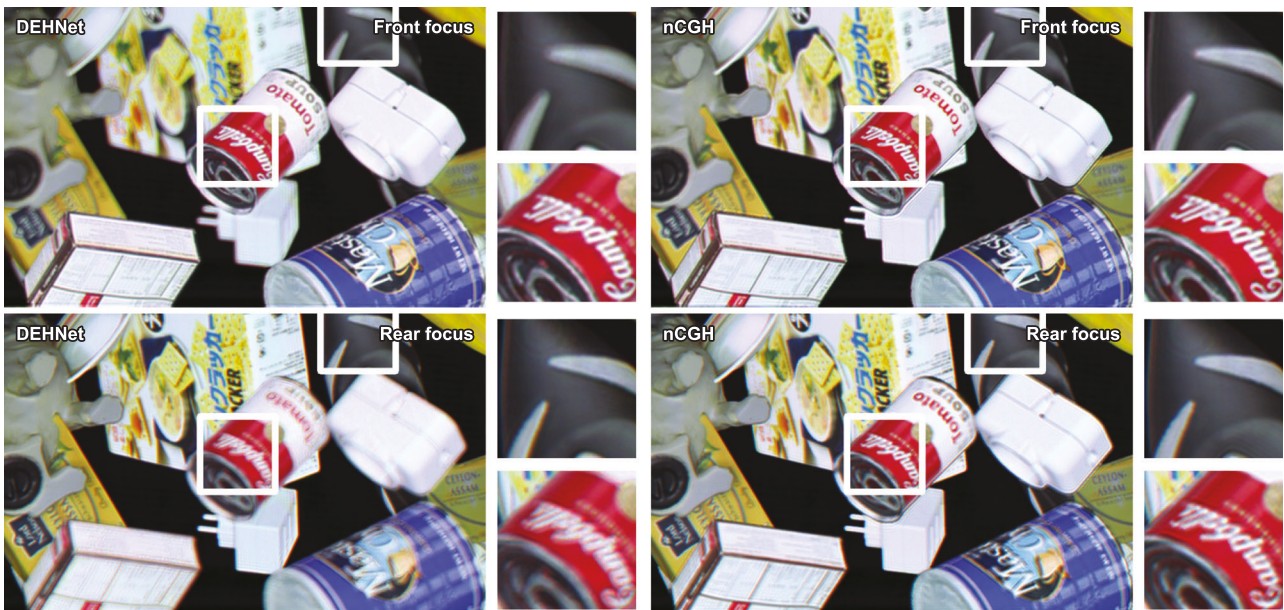

**Fig. 4 | Experimental results of the DEHNet and the nCGH.** The top images correspond to the front focus images and the bottom images correspond to the rear focus images. The small images represent enlargements of the corresponding reconstructions, as indicated by the white squares.

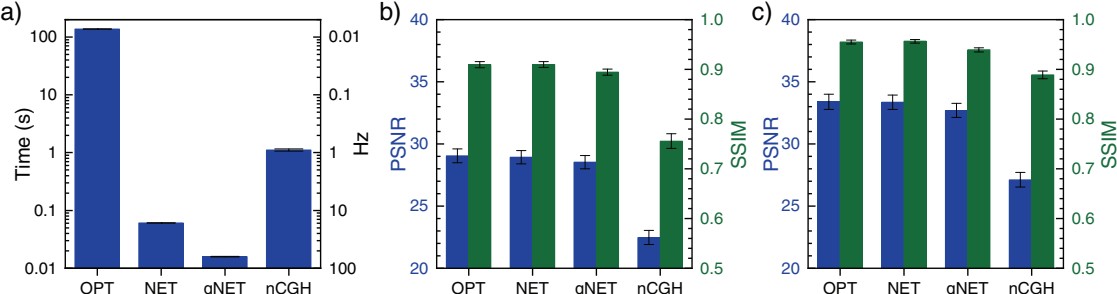

**Fig. 5 | Performance comparison. a** Inference times. OPT refers to the DEH calculated by optimization, NET refers to the DEH calculated by DEHNet before quantization, and qNET refers to the DEH calculated by DEHNet. We achieved a frame rate of 62 Hz in qNET, which is 8600 (70) times faster than that obtained in OPT (nCGH). The OPT inference time does not include the rendering time of the varifocal images. **b** The PSNR and SSIM were evaluated for the 512 resolution dataset. The presented PSNR and SSIM values represent the mean values of all images in the dataset with respect to the 21 rendered images for each scene. OPT, NET, and qNET have similar PSNR values (29.0, 28.9, and 28.5 dB, respectively) and SSIM values (0.909, 0.910, and 0.894, respectively), while nCGH gives significantly lower PSNR (22.5 dB) and SSIM (0.756) values. **c** The PSNR and SSIM were evaluated for the FHD resolution dataset. For the OPT, NET, qNET, and nCGH methods, the SSIM (PSNR) values were given by 0.955 (33.4 dB), 0.957 (33.4 dB), 0.939 (32.7 dB), and 0.889 (27.1 dB), respectively. The error bars represent the standard deviations between scenes.

The image quality, including defocus blur as well as speckle noise, can be measured quantitatively by evaluating the peak signal-to-noise ratio (PSNR) and the structural similarity (SSIM) compared to the rendered images. While the optimized DEH exhibits the best PSNR (26.1 dB) and SSIM (0.88) values, the DEHNet also gives compatible results. In contrast, the nCGH gives significantly lower PSNR (19.8 dB) and SSIM (0.67) values. Here, the second term in Eq. (2) boosts the image quality of the in-focus objects, which results in a slight reduction in the PSNR. Without the second term, the PSNR increases slightly (0.6 ~ 0.8 dB) although the image quality at the focal plane is reduced. The nCGH algorithm used here only includes ASM propagation and consideration of occluded surfaces, but a comparison with the other algorithm[16] can be found in Fig. S2. The other algorithm shows a similar weak defocus blur as the nCGH algorithm.

## Experiments and benchmark

In order to concretely validate the DEHNet, an experimental demonstration is necessary. In an optical reconstruction, an amplitude-only spatial light modulator (SLM) with a 1920 × 1080 (FHD) resolution is used instead of a complex SLM. It is well known that an amplitude SLM can be used as a complex SLM by means of spatial filtering, although the spatial bandwidth of the SLM is lost[37]. As confirmed by the simulation, the defocus blur is much weaker and a vivid boundary exists near the interface of the different-depth objects in the nCGH. As a consequence, it is difficult to perceive the depth of the 3D scene in the nCGH. This tendency is more apparent in the enlarged images shown in Fig. 4. Numerical and experimental reconstruction of nCGH and DEH with changing focal planes can be found in Supplementary videos. Details regarding the experimental setup and parameters can be found in the "Methods" section.

Figure 5 shows the inference times of the various CGH-generation methods, which were evaluated on an NVIDIA V100 GPU using the FHD resolution images. Since an optimization-based DEH requires 500 iterations, the method requires more than 1 min to synthesize a hologram with superior image quality. However, we achieved a frame rate of 62 Hz using DEHNet, while losing only ~0.5 dB of the PSNR

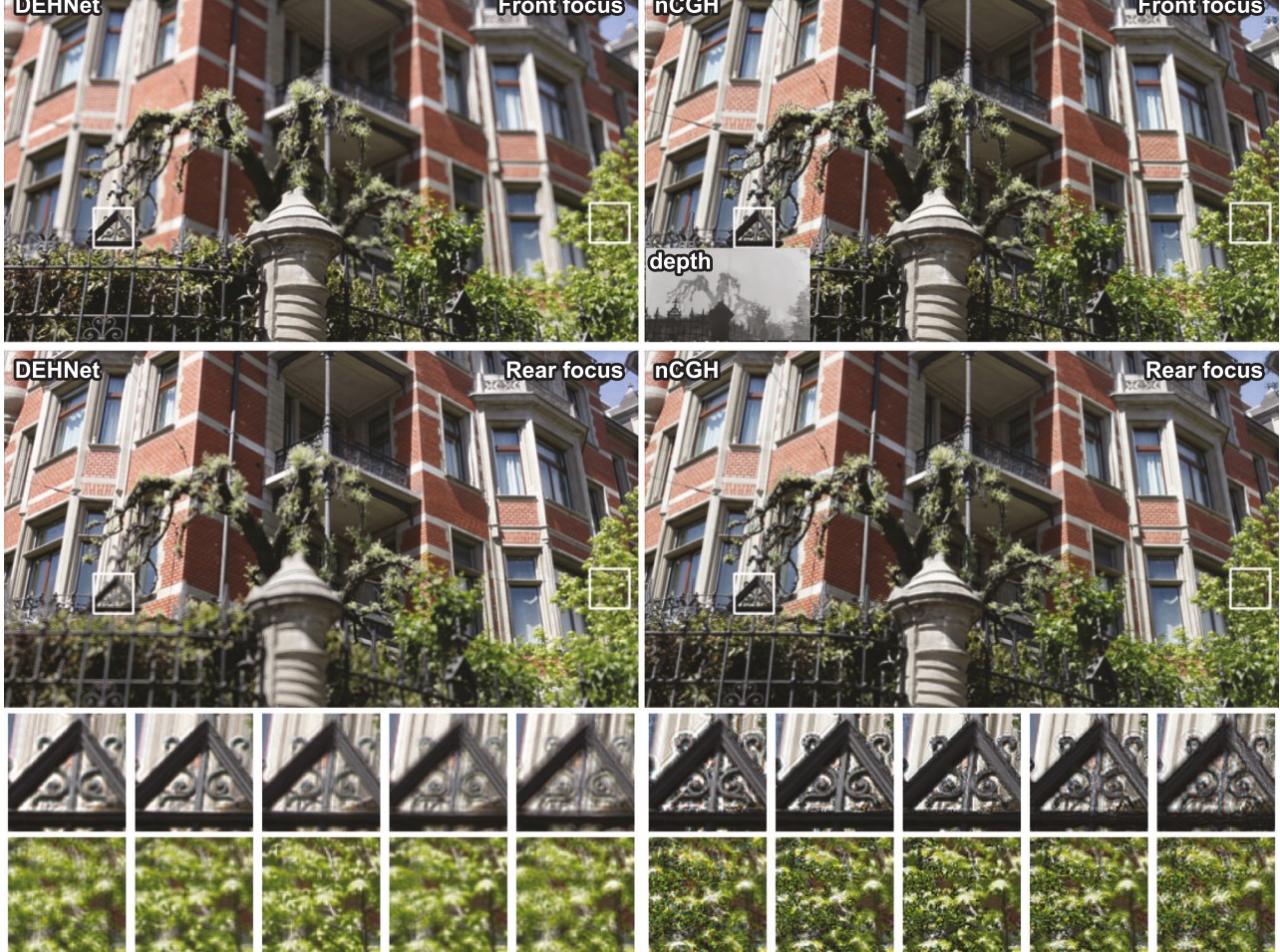

**Fig. 6 | Simulation results obtained using a real-world RGB-D image.** Using a real-world-captured image[56], a DEH and an nCGH were synthesized and their intensities were simulated wherein the total image size was resized to 1280 × 720. The large images show the front- and rear-focused images, while the small images show the focus-dependent images (from left to right, 0, 0.6, 1.2, 1.8, 2.4, and 3.0 diopter). The first row of small images shows the front objects and the rear objects simultaneously, while the second row of small images shows the noise on the leaves originating from the imperfect depth map.

compared to the optimization method. Considering encoding time, the total frame rate is 57 Hz since the encoding process of amplitude-only hologram takes 0.89 ms.

The PSNR and SSIM were evaluated for two datasets with different resolutions to quantitatively measure the image quality. One dataset is composed of 512 × 512 resolution images (Fig. 5b) as in the case of the training dataset, while the other dataset is composed of FHD resolution images (Fig. 5c). The indicated metrics represent the mean values of the comparison results between all varifocal images and the corresponding holograms so the smoothness of defocus blur and the sharpness of the focused object are both reflected. In the 512 (FHD) resolution dataset, the DEHNet provides a 6.5 (6.3) dB enhancement in the PSNR and a 0.15 (0.07) enhancement in the SSIM compared to the nCGH. Both of the evaluation datasets are rendered with textures that differ from that of the training dataset to ensure that the performance of the trained network is not restricted to the training dataset. Benchmark results with various image quality metrics including learned perceptual image patch similarity metrics[38] can be found in Section 2 of the Supplementary material.

When the holograms are synthesized using real-world images instead of rendered images, it should be pointed out that incorrect values from the captured depth maps can induce severe noise. In the majority of cases, real-world-captured depth maps include depth holes and incorrect depth values[39] so the interference pattern distorts the objects when the object boundaries of the depth map are not

consistent with those of the RGB image (Figs. 6 and 7). In contrast to the nCGH producing interference-induced black lines at the boundaries of noisy depth, the DEH provides noise-suppressed images at these boundaries. In some applications using measured depth maps, e.g. video see-through displays, the DEH would therefore give a superior image quality to the nCGH.

## Discussions

Recent advances in CGH algorithms result in remarkable progress in image quality and computation time. However, those algorithms do not give attention to the weak defocus blur of synthesized holograms, which severely distorts depth perception[22,23]. Because of the properties of static holograms, researchers believed that defocus blur would be correct in dynamic holographic displays. In this aspect, we demonstrate the difference between reconstructed images of the conventional holograms and realistic scenes depicting defocus blur depending on accommodation. Furthermore, we propose one solution to overcome the incorrect depth cue problem by adopting a neural network and the increments in the PSNR and SSIM metrics are substantial. We expect the DEHs could be widely used in holographic displays for virtual and augmented realities offering real-world-like 3D displays using currently available display devices.

Recently, several researchers reported enhanced accommodation of CGH by utilizing a number of the wave fields, the so-called time-multiplexing method[40,41]. Although the time-multiplexing method

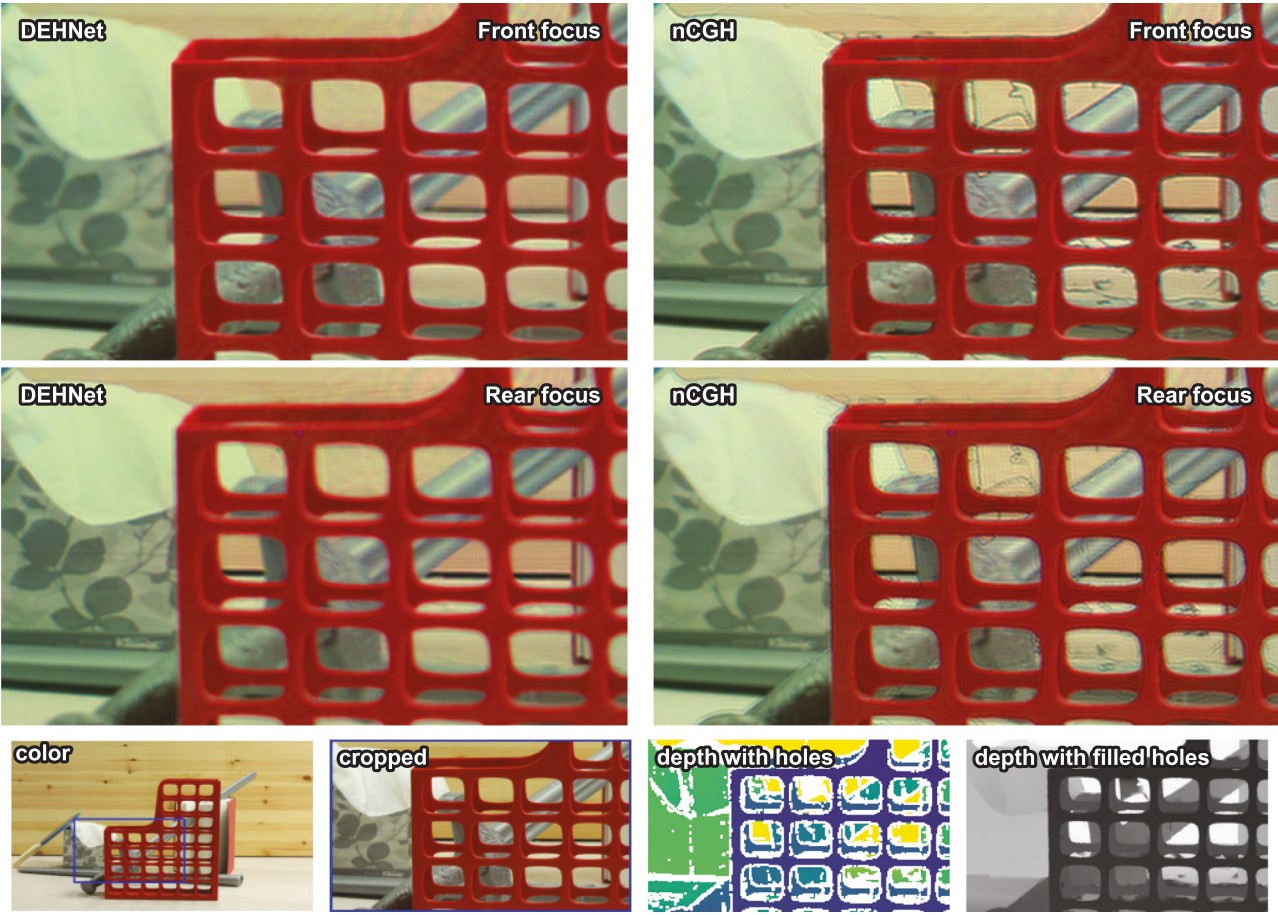

**Fig. 7 | Experimental results obtained using a real-world RGB-D image.** Using a real-world-captured RGB-D image[39], a DEH and an nCGH were synthesized and optically reconstructed. Since the real-world depth map includes depth holes, a monocular depth estimation algorithm[57] was adopted to fill the holes. The optically reconstructed images are cropped to show the details. Black lines caused by wave interference can be seen in the nCGH results but not in the DEH results. Moreover, defocus blur can be perceived only in the DEH results. "Color" represents the all-in-focus color image, "cropped" represents the cropped image, "depth with holes" represents the measured depth map depicting the depth holes with white color, and "depth with filled holes" represents the hole-filled depth map.

presents great image quality and enhanced accommodation effect, real-time reconstruction of such hologram requires more than 10 times of computation resources compared to our method and thus it is still challenging to realize real-time reconstruction. Besides, another approach is recently proposed to remove occlusion artifacts by adopting a layered depth image in learning-based CGH algorithms[42]. However, the approach did not deal with the amount of defocus blur, distinctive from our method.

It should also be noted here that in some works, the multi-plane hologram refers to the hologram reconstructing multiple objects at different depths, as an antonym of the hologram that reconstructs multiple objects at a single depth[36]. In contrast, we use the term to represent a hologram that can reconstruct numerous full-size images at the same time depending on the focal distance. As the latter hologram, our experiment shows a greatly enhanced image quality in comparison with that reported previously[30–33] despite the fact that more than 20 images were used as target images. The degraded image quality in previous experiments mainly originated from the high-frequency patterns that almost reached the pixel-pitch-limited frequency, since a phase-only SLM or an amplitude-only SLM was used instead of a complex SLM[30,34]. Our experiment confirms that it is possible to reconstruct multiple intensities with great fidelity when the target intensities are gradually varied, suggesting the feasibility of real-time applications of multi-plane holograms, such as holographic optical tweezers[32], one-step volumetric printings[43], and volumetric displays[44].

## Methods

### Determining the diameter of the blur circle

To construct large field of view (FoV) display systems, an SLM is magnified by a lens array. As a consequence, the maximum propagation distance of the hologram that allows the reconstruction of a virtual image with a depth from $d$ to infinity is determined by the effective focal length of the lens array. By approximating the lens array as a thin lens, the maximum propagation distance of the hologram, $\Delta z$, can be calculated as[45],

$$\Delta z \approx \frac{1}{d} \frac{\Delta x^2 \text{res}^2}{4 \tan^2 (\text{FoV}/2)}, \tag{3}$$

where $\Delta x$ is the pixel pitch of the SLM, res is the resolution of the display, FoV is the field of view of the system, $d$ is the virtual image distance of the floating object synthesized by the hologram, and the virtual image distance of the display is set to infinity. If we consider a 55° FoV, a 4K resolution, a 7.2 μm pixel pitch, and $d = 0.35$ m, then $\Delta z = 2$ mm is obtained from Eq. (3).

Under the specific display parameters that were considered herein, it is possible to calculate the diameter of a blur circle of a human eye when the eye is focused on infinity while the object synthesized by the hologram is floating at a distance of $d$. The diameter of a blur circle of an eye in units of display pixels, $\text{CoC}_{\text{eye}}$, is given by

$$\text{CoC}_{\text{eye}} = \frac{A \cdot \text{res}}{2d \cdot \tan(\text{FoV}/2)}, \tag{4}$$

where $A$ is the pupil diameter. If the wave field of the hologram is partially blocked by the iris, the image quality degrades by the noise of the blocked wave field. Considering that the diameter of a pupil is larger than 1.5 mm in the majority of cases[46,47], $A$ is set to 1.5 mm to avoid image degradation originating from a partially blocked wave field. From the above parameters, $CoC_{eye}$ is 15 pixels and the aperture size of the rendering camera is set to satisfy the diameter of a blur circle of the rendered images.

Although the diameter of defocus blur of an nCGH can be enlarged by increasing the propagation distance, achieving a blur circle equivalent to that of a human eye is only possible under a small FoV (-10°). For example, if we increase the propagation distance to enlarge the diameter of the blur circle, the virtual image distance of the object($d$) comes closer and the blur circle diameter of the eye ($CoC_{eye}$) is also increased. As a result, an increase in the diameter of the defocus blur under a fixed propagation distance is required to attain a human eye-equivalent defocus blur with a holographic display.

## Experimental details

In the experiment, IRIS-U62 LCoS (liquid crystal on silicon) from MAY Inc. of which resolution is 3840 × 2160 and pixel pitch is 3.6 μm, is used as 1080p mode by putting the same pixel value in 2 × 2 nearest pixels to minimize pixel crosstalk originated from its small pixel pitch[48]. As a result, the LCoS behaves as an FHD resolution amplitude-only LCoS with a pixel pitch of 7.2 μm. The distance between the minimum and maximum depths was set to 2 mm. The dispersion diameters by the pixel pitch diffraction are 25 (red), 20 (green), and 18 pixels (blue) under 2 mm light propagation. Considering that the maximum diameter of defocus blur of the rendered images is 15 pixels, the propagation distance should be longer than 1.7 mm. Since the modulated intensity nonlinearly depends on the assigned values of the pixels, the amplitude was calibrated by measuring output values for each input pixel value. An off-axis hologram was adopted and the grating period was set to 0.25 of its maximum period to avoid unwanted noise. The Burch encoding method[49] was used to project the complex wave field onto real values. With an adjustable 2D slit, zeroth order and higher order diffractions are blocked (Fig. 8). As a light source, laser diodes with wavelengths of 638, 515, and 460 nm were used and were sequentially illuminated on the LCoS. To remove speckles caused by the coherence of the lasers, the holographic diffuser was rotated at the focused spot of the laser beams.

## Phase noise of the amplitude-only SLM

Due to the properties of liquid crystals, it is inevitable that the amplitude-only SLM modulates the phase. The noise from such phase modulation can be avoided if an appropriate grating phase is applied. Assuming that amplitude modulation is given by $f(x)$ and unwanted phase modulation is given by $\exp\{ip_1 f(x) + ip_2 f(x)^2\}$, then the wave field at the SLM is given as $f(x)e^{ip_1 f(x) + ip_2 f(x)^2}$. Here, we approximated the unwanted phase modulation as a second-order polynomial function of the amplitude modulation. To expand the expression, we employed the Jacobi–Angler expansion, $e^{ikz\cos\theta} = \sum_{n=-\infty}^{\infty} i^n J_n(z)e^{in\theta}$, where $J_n(z)$ is the $n$th Bessel function of the first kind. Using a Fourier series expansion, $f(x) = \sum_k F_k \cos(kx + \phi_k)$, the wave field at the SLM can be expressed as

$$
\begin{aligned}
f(x)e^{ip_1 f(x) + ip_2 f(x)^2} &= f(x)e^{ip_1\left(\sum_k F_k \cos(kx+\phi_k)\right) + ip_2\left(\sum_k F_k \cos(kx+\phi_k)\right)^2} \\
&= f(x) \prod_k \sum_n i^n J_n(p_1 F_k)e^{in(kx+\phi_k)} \\
&\quad \times \prod_{k,l} \sum_n i^n J_n(p_2 F_k F_l/2)e^{in((k+l)x+\phi_k+\phi_l)} \\
&\quad \times \prod_{k,l} \sum_n i^n J_n(p_2 F_k F_l/2)e^{in((k-l)x+\phi_k-\phi_l)}.
\end{aligned} \tag{5}
$$

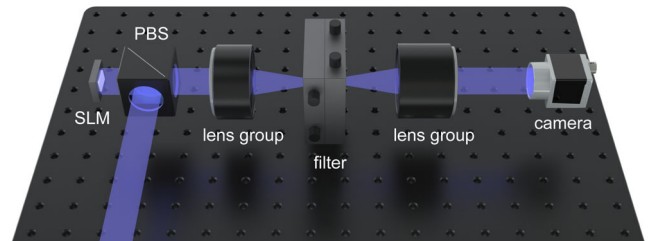

**Fig. 8 | Schematic representation of the experimental setup.** Collimated RGB lasers were illuminated on an SLM through a polarizing beam splitter (PBS). After the Fourier plane is formed by the first lens group, zeroth order and higher orders of the grating phase diffraction implemented on the SLM were blocked by the filter. The reconstructed hologram was captured by a camera with a second lens group.

Fortunately, $p_1$, $p_2$, and $F_k$ are <1 in our experiment, and so $J_n(z)$ with $|n| \ll 1$ can be neglected for those cases. As a result, Eq. (5) can be approximated as

$$
\begin{aligned}
f(x)e^{ip_1 f(x) + ip_2 f(x)^2} &\approx f(x) \prod_k J_0(p_1 F_k)\left(\prod_{k,l} J_0(p_2 F_k F_l/2)\right)^2 \times \Bigg[ \sum_m \frac{iJ_1(p_1 F_m)}{J_0(p_1 F_m)}e^{i(mx+\phi_m)} \\
&\quad + \sum_{m,n} \frac{2iJ_1(p_2 F_m F_n/2)}{J_0(p_2 F_m F_n/2)}\left(e^{i((m+n)x+\phi_m+\phi_n)} + e^{i((m-n)x+\phi_m-\phi_n)}\right) \\
&\quad + \mathcal{O}\left((p_1 F_k)^2\right) + \mathcal{O}\left((p_2 F_k^2)^2\right) \Bigg].
\end{aligned} \tag{6}
$$

As we can see from Eq. (6), if a grating phase with a period $e^{ik_{prism}x}$ is applied, then $e^{ik_{prism}x}$, $e^{-ik_{prism}x}$, $e^{2ik_{prism}x}$, and a constant term is generated. Moreover, Burch encoding[49] generates its conjugate term $e^{-ik_{prism}x}$ and its phase noise-induced terms. As a result, the $e^{ik_{prism}x}$, $e^{-ik_{prism}x}$, $e^{-ik_{pitch}x+2ik_{prism}x}$, $e^{-ik_{prism}x}$, $e^{ik_{prism}x}$, $e^{ik_{pitch}x-2ik_{prism}x}$ terms exist, where $k_{pitch}$ is the wavenumber of the SLM pixel pitch and the terms such as $e^{ik_{pitch}x-2ik_{prism}x}$ are created by the black matrix of the SLM. When the frequency of the grating phase is one-third of the spatial frequency of the pixel pitch, our signal term $e^{ik_{prism}x}$ overlaps with the noise term $e^{ik_{pitch}x-2ik_{prism}x}$ and the noise cannot be filtered. To avoid such noise, the frequency of the grating phase was set to one-quarter or less of the spatial frequency of the pixel pitch.

## Generation of the training dataset

The objects in the 3D scene were randomly sampled from publicly available datasets[50–53] and each scene was rendered by Blender to have 21 varifocal images[54]. The textures of the objects used in the training stage were randomly sampled from the CC0 texture library[55] and the textures of the objects used in the evaluation stage were sampled from the "Benchmark for 6D Object Pose Estimation" datasets[50–53]. The colors, orientations, and intensities of the light sources were randomly sampled while the maximum intensity was restricted to prevent overexposure. When a scene is overexposed, intensity sums of each varifocal image could be different because the intensities become clipped. Since the propagation of light conserves its total energy, varifocal images with inconsistent intensity sums cannot be constructed with a single wave field.

The focal planes of each scene were equally spaced while the distances between the camera and the objects were significantly longer than the distances between the different objects to symmetrically blur either side of the focal plane. The symmetric blur in the rendered images is consistent with the asymmetric blur of an eye when a tiny display is magnified and projected to the eye. With the exception of the background, the pixel-wise statistics of the depth distribution were made almost uniform to prevent overfitting to a particular depth

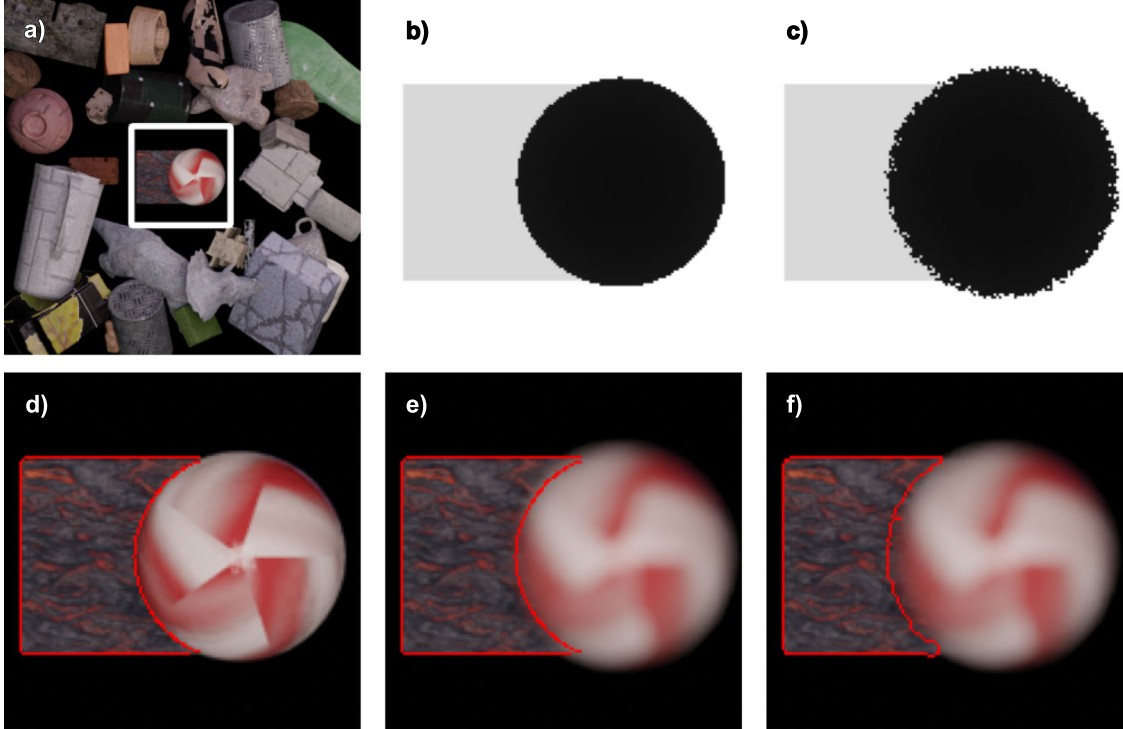

**Fig. 9 | All-in-focus depth map and defocus-blur-considered depth map overlaid on a color image. a** All-in-focus rendered image. White box indicates an enlarged area for other subfigures. **b** Enlarged all-in-focus depth map. **c** Defocus-blur-considered depth map for the rear plane of focus. While acquiring the depth values in the area of the defocus blur, the depth values of the front object and that of the rear object were randomly sampled by the renderer. To ignore the depth values of the rear object near the boundary, the depth maps were acquired multiple times and the most front values among the numerous depth maps were used. **d** The boundary of the rear object of the all-in-focus depth map is marked as a red line in the rendered all-in-focus image. **e** The boundary of the rear object of the all-in-focus depth map is marked as a red line in the rendered rear-focus image. **f** The boundary of the rear object of the rear-focus depth map is marked as a red line in the rendered rear-focus image.

during training (see Section 7 of the Supplementary material for further details of the depth distribution).

## Parameters of the loss function and the depth map with defocus blur

Since the objects in the scene can have any depth, the number of varifocal images was selected to be 21 pixels larger than the maximum diameter of the blur circle, while $\gamma$ was fixed to 40 to avoid the simultaneous focusing of an object at two different focal planes. For an arbitrary object, the number of out-of-focus images (20) is significantly larger than the number of in-focus images (1) and so the reconstructed scene of the DEH is more influenced by the blurred images than the focused image. Thus, to apply a similar or higher weight to an in-focus image of objects, $\beta$ was set to 20.

Although defocus blur is not considered while synthesizing a depth map in the majority of applications, we used a defocus-blur-considered depth map during the optimization and training processes to consider occluded surfaces. Normally, if we include defocusing blur when rendering a depth map, front depth values and rear depth values are blended at an edge of defocus blur of a front object. Instead, we sampled the depth map using only one ray per pixel and collected 10 depth maps for each focal distance. Among 10 depth values of each pixel, only the front-most depth value is used, so the depth values of the blurred pixels are confined to the depth of the front object. If we assume that one object is located at the front of the scene and another object is located at the rear of the scene, a blur circle of the rear object does not invade a focused image of the front object when the front object is focused. In contrast, a blur circle of the front object invades a focused image of the rear object when the rear object is focused (Fig. 9). Assuming

that an all-in-focus depth map is used when comparing the depth-weighted all-in-focus image and the intensity of the hologram for the rear plane of focus (second term of Eq. (2)), the pixel weights of the rear object close to the front object are high even if the blur circle degrades the image quality. As a result, the loss function has a lower value when a sharply focused image is reconstructed near the boundary of the front object, ignoring the defocus blur of the front object. Such circumstances can be avoided when the defocus-blur-considered depth map is used for the second term of Eq. (2) since the rear object occupies a smaller area in this depth map than in the all-in-focus depth map for the rear plane of focus.

In an aspect of the loss function, we tried to adopt multi-scale structural similarity (MS-SSIM) loss instead of MSE loss while synthesizing DEH. However, the effect of MS-SSIM loss was unclear and the DEH optimized by MS-SSIM loss suffered from defects. Comparison results can be found in Section 12 of the Supplementary material.

## Training of the neural network

In the first stage of training, we used batch normalization layers in front of activation layers. When the validation loss stopped decreasing, the batch normalization layers and convolution layers were manually fused using running means and running variances. After fusing the batch normalization layers and convolution layers, the fused layers were trained again with the same dataset until the validation loss stopped decreasing. We used the Adam optimizer with a learning rate of 0.0005. We reset the internal parameters of the optimizer for every 50 epochs at the second stage of training. The batch size was 16 and the weights of the network are updated after 4 batch runs, yielding an effective batch size of 64. The training process took approximately

60 h using an NVIDIA V100 GPU. The trained neural network was symmetrically quantized using the TensorRT library and the same training dataset was fed to calibrate the quantization parameters.

## Data availability

All relevant data that support the findings of this work are available from the corresponding author upon reasonable request. The configuration settings of BlenderProc used in synthesizing the training and evaluation datasets will be publicly available along with the paper.

## Code availability

All relevant codes that support the findings of this work are available from the corresponding author upon reasonable request.

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

## Author contributions

D.Y. conceived the idea and wrote the manuscript. W.S. and H.Y. were involved in developing the proposed algorithm. D.Y. performed the experiments with help from W.S., S.I.K., B.S., C.-K. L., and S.M. H.Y., J.K., J.-Y.H., and G.S. contributed to the theoretical investigations. H.-S.L. supervised overall work. All authors participated in discussions and contributed to the manuscript.

## Competing interests

The authors declare no competing interests.
