## [Peer review file · Nature Communications]

REVIEWER COMMENTS

Reviewer #1 (Remarks to the Author):

This manuscript proposes a hologram that imitates defocus blur of incoherent light by engineering diffracted pattern of coherent light, called diffraction-engineered hologram (DEH). The DEH employed an advanced physically-based renderer to generate tens of varifocal images of the target scene. The rendered images have accurate occlusion and blur circles. They are used to optimize the phase distribution of the target wave field by the iterations. The optimized wave field has a clear display effect on the object-existing planes and naturalistic blur on the out-of-focus planes. And then it is propagated by the ASM method to generate the hologram for display. A network-based synthetic method is demonstrated to speed up the calculation, called DEHNet. The optical reconstructions with more accurate blur circles can be real-timely obtained.

This manuscript employs several available algorithms for the CGH generation, such as the gradient descent method, the angular-spectrum method, and the amplitude-only hologram conversion algorithm. Its major novelty is the naturalistic blur display effect based on the advanced renderer. The naturalistic blur challenge is common in CGH displays. And the authors first propose to solve it by rendering, relying on the computing resources. It is a new feasible idea and preliminary attempt. The manuscript is well-written and the results should be interesting to the readers. The revision suggestions are as follows.

1. The definition of diffraction-engineered hologram (DEH) is not clear as the main body of this manuscript. I am still confused about this after reading the whole manuscript. It is my understanding that the DEH is the optimized complex amplitude distribution at the $z=0$ plane, and the propagation distance for display is the average focal distance of the varifocal images. It should be clearly defined and marked in Fig. 1c for an enhanced presentation.
2. The “corresponding holograms” in the caption of Fig. 3 should be “corresponding reconstructions”.
3. From the depth map in Fig. 2, the objects seem to be distributed in just two layers. I doubt this can limit the generalization the network for real scenes with continuous depth contribution, like Figure Extended 2. Is it necessary to carry out the transfer learning for it?
4. How long does it take to convert the DEH to the amplitude-only hologram for experimental display? What is the frame rate including the conversion time?

5. The authors argued that “The diffusive hologram is excluded from the comparison because its image quality is not compatible with other methods unless other techniques, such as the time-multiplexing technique, are adopted simultaneously” in line 161. It is suggested to provide more explanation. As far as I know, lots of dCGH works have been proposed for 3D display, like Ref. 16.

6. To make it easy to follow, it is suggested to mark the “In-focus intensities” and “Out-of-focus intensities” in Fig. 1e. And it is too small to see the differences between the in-focus intensities of the dCGH, nCGH, and DEH.

7. The differences between the enlargements in Fig. 3 are not obvious. It is suggested to reselect the enlarged areas, like the red biscuit box and the white stopper beside it.

8. The commas of Eq. M1 and M2 are lost.

Reviewer #2 (Remarks to the Author):

*** Paper summary ****

1) The paper introduces an innovative way of optimizing complex holograms for more accurate reproduction of the depth of field effect. Similarly to previous methods, it optimizes the hologram image to minimize difference between the simulated hologram intensity at different planes and the target focal stack. Differently from previous methods, the authors do not use wave propagation but off-the-shelf ray tracer to produce the target focal stack. This leads to a visually more pleasing defocus appearance. Furthermore, the authors do not enforce smooth phase allocation which gives which allows for a more accurate reconstruction.

*** Justification of rating ***

2) Overall, the paper is very well written and easy to follow. There is minimum of typos and technical issues. The structure is clear and well balanced. The main weakness of the submission is the relatively minor technical novelty. While the overall complexity of the work is non-trivial, the method is largely similar to previous methods that optimize against a focal stack and the main innovation is in the way the ground truth focal stack has been obtained. Despite that, the work is well crafted and it does achieve its

objective based on the relatively complete set of evaluation benchmarks. Therefore, I am quite hopeful the method will be accepted after a revision round to clarify some of the questions that follow.

*** Questions and comments ***

* Exposition *

2) The placement of reference labels is somewhat unpolished with spaces often missing: "on.[22,23]"

3) The method suddenly introduces concept of voxels without clarifying what this refers to (L43). Typically, voxels would be an element of a discretized volume but no such concept has been suggested here. Furthermore, how does one propagate a voxel (L135)?

4) Minor issues:

- L151: "Non-linearity and a wide receptive field are more important than hidden features so the number of channels" - How has this been determined/validated?

- L198: "Although it was not mentioned" - why not specify it directly in the exposition then?

- The PSNR values should include units (dB).

* Related work *

5) The the extent of my knowledge the authors discuss relevant work. Additionally, the authors should cite another recent method for high-quality 3D holography:

Lee, Byounghyo, et al. "High-contrast, speckle-free, true 3D holography via binary CGH optimization." Scientific reports 12.1 (2022): 1-12.

This work also considers focal stack when optimizing the target hologram. The authors should discuss how different their approach is.

* Reproducibility *

6) The authors have provided detailed description and assessment of their system. What I could not find is exact type of their SLM. I also miss such level of description for the exact architecture of their network and the training curriculum (optimizer, learning rate,...). The authors could also clearly indicate what part of the data will be shared with other researchers (if any).

* Technical aspects and validation *

7) What is the purpose of the explicit all-in-focus loss component? The objects at focal plane should already be focused in the respective focal stack layers so why the special treatment? It seems to cause additional complications as discussed on L334. I miss any study of the behavior that this attempts to mitigate (e.g. ablation).

8) Unlike in some of the other methods [15,16], the authors do not force the phase field to be smooth. They demonstrate that this does not pose significant issues for their amplitude-only prototype but I wonder if this would lead to increased artifacts for a phase-only SLM where a suitable coding strategy had to be used (e.g. double phase encoding).

9) The captured images presented in Figure 4 do not exhibit as excellent defocus quality as the simulated images. It is my understanding that the (almost) PSF in Figure 1f has been simulated. Would it be possible to provide the same PSF comparison captured on a real setup? It is not clear how the theoretical benefit of arbitrarily optimized phase transfers to an amplitude-encoded and physically constrained real display device.

10) How can the method account for light contribution from scene points occluded in the pinhole RGBD input image?

11) The PSNR is most certainly not a "perceptual image quality" metric (L172). The authors could consider adding a perceptually validated metric such as LPIPS.

12) Which nCGH method has been used in the comparisons? The timings in Figure 4 seems to assume no neural acceleration has been available for the nCGH, yet previous work has demonstrated that such approach is possible and leads to real-time performance [16]. Were the comparisons done using similarly scaled approaches?

13) How were the defocus-blurred depth maps obtained (L335)? The Extended Figure 4 suggests that some form of random sampling has been utilized but the application of said depth maps suggest that a soft depth gradient is intended. Have the maps been additionally filtered to that goal?

14) Was the method [1] in the supplement trained on the same data as the proposed method?

Reviewer #3 (Remarks to the Author):

The authors present a method to generate multiplane holograms of 3D color scenes with improved depth perception and non-diffuse phase. In particular, their method solves one of the main issues with non-diffuse phase holograms, namely the lack of natural blur in the defocused regions of the reconstructed scene. To achieve this, they make use of a physically-based renderer that simulates the adequate defocus blur at different depths of the scene and then feeds this render and the color and depth information of the scene to a convolutional neural network, which is trained to generate the adequate field distribution. Once the neural network is trained, only the color and depth map of a scene is required to generate its hologram. The presented results are remarkable, dealing with a relatively unexplored issue, and the paper is well written. In particular, adequate depth blur is reproduced at different depths after reconstructing the generated hologram, compared to the lack of such blur in common in conventional multiplane hologram generation. This has the potential to lead to improvements in holographic displays and associated applications.

Nevertheless, some important points should be addressed.

1. The results in this paper use 21 varifocal images to generate the holograms, however, using as a criterion that “the number of varifocal images was selected to be 21 pixels larger than the maximum diameter of the blur circle”. I believe this point is essential to understanding the performance of the method, and as such the authors should expand the associated discussion. Some relevant questions are
 - How many individual planes must be rendered to achieve a hologram with optimal depth blur?
 - Does the number of planes depend on the target depth of the scene?
 - Is the number of individual planes fixed after training the neural network?
 - Which is the maximum scene depth that can be successfully reconstructed with the proposed method?
 - In conventional multiplane holograms, the computation time has a strong relationship with the number of individual planes. Is this the case with this approach?

2. Did the authors measure if there is any difference in the diffraction efficiency of the holograms generated with gradient descent and the neural network? This is particularly important since the use of amplitude-only SLM already implies a lower diffraction efficiency compared to phase-only modulation.

3. The introduction lacks an overview of the recent work in multiplane hologram generation. Remarkably, there is only a single recent citation regarding this point [26], and most are from 2010 and before. The authors should consider a careful review of the recent literature to update the bibliography.

Some relevant references

[1] C. Chen, B. Lee, N.-N. Li, M. Chae, D. Wang, Q.-H. Wang, and B. Lee, "Multi-depth hologram generation using stochastic gradient descent algorithm with complex loss function," *Opt. Express* 29, 15089–15103 (2021)

[2] J. Zhang, N. Pégard, J. Zhong, H. Adesnik, and L. Waller, "3D computer-generated holography by non-convex optimization," *Optica* 4, 1306–1313 (2017)

[3] P. Zhou, Y. Li, S. Liu, and Y. Su, "Dynamic compensatory Gerchberg–Saxton algorithm for multiple-plane reconstruction in holographic displays," *Opt. Express* 27, 8958–67 (2019)

[4] A. Velez-Zea, J. Fredy Barrera-Ramírez, and R. Torroba, "Improved phase hologram generation of multiple 3D objects," *Appl. Opt.* 61, 3230 (2022)

[5] C. Ying, P. Hui, C. Fan, and W. Zhou, "New method for the design of a phase-only computer hologram for multiplane reconstruction," *Opt. Eng.* 50, 055802 (2011)

4. The loss function used in this work is a per-pixel error metric, however, this is not necessarily the best approach when improved perception of the hologram reconstruction is desired. An alternative better suited for human perception is the structural similarity index (SSIM). Did the authors test or consider alternative loss functions? Some discussion regarding this point would clarify this point. A relevant reference is Yang, F., Kadis, A., Mouthaan, R. et al. Perceptually motivated loss functions for computer-generated holographic displays. *Sci Rep* 12, 7709 (2022). <https://doi.org/10.1038/s41598-022-11373-8>

5. Figure 1 is extremely important in the present manuscript, showing how the different methods deal with the phase information, the rendering process, the DEH method, the convolutional neural network, and the in-focus and out-of-focus intensities in each method. Both the figure and caption are extremely dense, and the font size in the figure is very small, making its reading and interpretation difficult. I believe that each of these points merits an individual figure with their corresponding discussion and caption, or at least a larger size for each subfigure.

6. The labels in figure 2 are difficult to read due to the poor contrast with the object. This should be addressed, for example by using a black outline for the labels. Consider also using letter labels to refer to each figure instead of their position (bottom-left, top-left). This would make the caption easier to read.

7. The indents of the zoomed-in regions showing the depth effects in Figures 2, 3, and the extended figures 1 and 2 should be larger to allow a better perception of the difference between the methods. This should also be done in the figures in the supplementary material, in particular, figure S5 lacks zoomed indents that allow perceiving the effect.

8. The authors should provide the model of the SLM used in the experiment.

9. To better show the effect of the depth blur, the authors should consider including a video of both the numerical and experimental reconstruction of one of the scenes as the focus plane is changed.

Given the above points, I believe this paper can only be considered for publication after major revisions.

Reviewer #1 (Remarks to the Author):

This manuscript proposes a hologram that imitates defocus blur of incoherent light by engineering diffracted pattern of coherent light, called diffraction-engineered hologram (DEH). The DEH employed an advanced physically-based renderer to generate tens of varifocal images of the target scene. The rendered images have accurate occlusion and blur circles. They are used to optimize the phase distribution of the target wave field by the iterations. The optimized wave field has a clear display effect on the object-existing planes and naturalistic blur on the out-of-focus planes. And then it is propagated by the ASM method to generate the hologram for display. A network-based synthetic method is demonstrated to speed up the calculation, called DEHNet. The optical reconstructions with more accurate blur circles can be real-timely obtained.

This manuscript employs several available algorithms for the CGH generation, such as the gradient descent method, the angular-spectrum method, and the amplitude-only hologram conversion algorithm. Its major novelty is the naturalistic blur display effect based on the advanced renderer. The naturalistic blur challenge is common in CGH displays. And the authors first propose to solve it by rendering, relying on the computing resources. It is a new feasible idea and preliminary attempt. The manuscript is well-written and the results should be interesting to the readers. The revision suggestions are as follows.

=> Before we respond to Reviewer 1's questions and comments, we would like to thank Reviewer 1 for making helpful comments and important questions. In particular, we would like to thank Reviewer 1 for bringing up unclearly stated contents in our manuscript. After revising the manuscript following Reviewer 1's comments, the manuscript becomes clearer than before.

1. The definition of diffraction-engineered hologram (DEH) is not clear as the main body of this manuscript. I am still confused about this after reading the whole manuscript. It is my understanding that the DEH is the optimized complex amplitude distribution at the $z=0$ plane, and the propagation distance for display is the average focal distance of the varifocal images. It should be clearly defined and marked in Fig. 1c for a enhanced presentation.

=> We appreciate Reviewer 1 for pointing out important point and we apologize for our mistake. As Reviewer 1 indicated, we optimized complex amplitude distribution at the $z=0$ plane. However, the sentence "the wave field is subsequently propagated by the average focal distance of the varifocal images." does not reflect the change of the algorithm. While improving the algorithm to speed up, we confirmed the last propagation is not required. Instead, a virtual image of the SLM is formed by the optical system and the virtual image meets the focal distance condition of the rendered images (Figure S4). Although we changed Figure 1c and other sentences, we didn't delete the part of the sentence, "the wave field is subsequently propagated by the average focal distance of the varifocal images", by mistake. We removed the part from the sentence to clearly present it. Moreover, we marked "propagate & compare with varifocal images" in Fig. 1c to clearly explain the algorithm. Again, we appreciate for pointing out our mistake.

The following figure is revised version of Fig. 1c.

2. The “corresponding holograms” in the caption of Fig. 3 should be “corresponding reconstructions”.

=> The words are changed as Reviewer 1 commented. We thank Reviewer 1 for kind advice.

3. From the depth map in Fig. 2, the objects seem to be distributed in just two layers. I doubt this can limit the generalization the network for real scenes with continuous depth contribution, like Figure Extended 2. Is it necessary to carry out the transfer learning for it?

=> In Fig. 2, to clearly present the effect of our method, we synthesized a scene of which depth differences between objects are significant. Although the depth distribution of the scene used in Fig. 2 seems to have only 2 depth layers, there exist other objects with other depth. For instance, a plate at bottom-left corner is located at different depth.

For the training, scenes in training dataset are composed of objects with various depths and the depth distribution of training dataset is nearly uniform. The following figure presents average depth distribution of all scenes in the training dataset. Each bar represents depth probability between the range $[x-0.05, x+0.05)$, where x refers a labeled value in the figure. The depth value 0 corresponds to the minimum depth and the depth value 1 corresponds to the maximum depth. BG refers background of scenes, corresponding to the maximum depth value. Error bar represents standard deviation of each distribution between scenes. Depth distribution near the maximum depth is minimized to overlap between objects and the background.

To accomplish such distribution, objects are moved before rendering scenes if some depth range is too crowded. During re-distributing the objects, we divided the whole depth range into 5 ranges and moved 3 objects at the depth with maximum probability if the maximum depth probability is 20% larger than the minimum depth probability.

The following figure and the related explanation are added to the supplementary material.

4. How long does it take to convert the DEH to the amplitude-only hologram for experimental display? What is the frame rate including the conversion time?

=> For amplitude-only hologram, encoding of DEH takes 0.89 ms and thus the frame rate is around 57 Hz. We added the following sentence in the manuscript.

With considering encoding time, the total frame rate is 57 Hz since the encoding process of amplitude-only hologram takes 0.89 ms.

5. The authors argued that “The diffusive hologram is excluded from the comparison because its image quality is not compatible with other methods unless other techniques, such as the time-multiplexing technique, are adopted simultaneously” in line 161. It is suggested to provide more explanation. As far as I know, lots of dCGH works have been proposed for 3D display, like Ref. 16.

=> Here, we used the word, diffusive hologram, as a hologram which scatters light to wide angle. Since scattered lights come into a human eye in various directions, defocus blur of dCGH should be similar to that of real-world scenes. However, as far as we know, most of the papers with high image quality do not scatter lights since scattering lights into various directions degrades image quality. We classified Ref. 16 as nCGH in a perspective of light-scattering since the paper utilized smooth phase.

For more concise comparison, we compared Ref. 16 (representative example of nCGH), Ref. 12 (representative example of dCGH), and DEH. The following figure presents reconstructed images of different algorithms and different focal planes. Tensorholo refers Ref. 16 and focal planes of the represented images correspond to 0.15, 1.05, 1.95, and 2.85 diopter (from left column to right column). In our opinion, smooth phase of Ref. 16 prevent scattering of light and thus defocus blur is weak.

Moreover, the wording, “because its image quality is not compatible with other methods”, seems offensive so we changed the sentence to “We only compared non-diffusive holograms in this paper because the purpose of the paper is synthesizing high-image-quality holograms. Comparison between diffusive hologram and DEH can be found in the supplementary material.” Also, we included the above figure in the supplementary material.

*Ref [16]: Maimone, A., Georgiou, A. & Kollin, J. S. "Holographic near-eye displays for virtual and augmented reality." *ACM Transactions on Graphics (Tog)* **36**, 1–16 (2017).

6. To make it easy to follow, it is suggested to mark the "In-focus intensities" and "Out-of-focus intensities" in Fig. 1e. And it is too small to see the differences between the in-focus intensities of the dCGH, nCGH, and DEH.

=> We marked "In-focus intensities" and "Out-of-focus intensities" in Fig. 1e as Reviewer 1 suggested. To increase the size of Fig. 1e., we divide Fig 1. into 2 different figures and increased the size of Fig. 1e.

7. The differences between the enlargements in Fig. 3 are not obvious. It is suggested to reselect the enlarged areas, like the red biscuit box and the white stopper beside it.

=> We replaced Fig. 3 to emphasize the difference. We moved objects to the maximum and minimum depth of the scene and did experiment with it. Moreover, we cropped the captured image and reselected enlarged area to present defocus blur. (Please see the following figure)

8. The commas of Eq. M1 and M2 are lost.

=> We thank Reviewer 1 for the suggestion of typos. We corrected the typos.

Reviewer #2 (Remarks to the Author):

*** Paper summary ***

1) The paper introduces an innovative way of optimizing complex holograms for more accurate reproduction of the depth of field effect. Similarly to previous methods, it optimizes the hologram image to minimize difference between the simulated hologram intensity at different planes and the target focal stack. Differently from previous methods, the authors do not use wave propagation but off-the-shelf ray tracer to produce the target focal stack. This leads to a visually more pleasing defocus appearance. Furthermore, the authors do not enforce smooth phase allocation which gives which allows for a more accurate reconstruction.

*** Justification of rating ***

2) Overall, the paper is very well written and easy to follow. There is minimum of typos and technical issues. The structure is clear and well balanced. The main weakness of the submission is the relatively minor technical novelty. While the overall complexity of the work is non-trivial, the method is largely similar to previous methods that optimize against a focal stack and the main innovation is in the way the ground truth focal stack has been obtained. Despite that, the work is well crafted and it does achieve its objective based on the relatively complete set of evaluation benchmarks. Therefore, I am quite hopeful the method will be accepted after a revision round to clarify some of the questions that follow.

=> Before we respond to Reviewer 2's questions and comments, we would like to thank Reviewer 2 for making valuable comments and efforts towards improving our manuscript. Thanks to helpful comments of Reviewer 2, we can improve our manuscript by including new sections we missed before. We hope our revised manuscript can clearly present our idea.

*** Questions and comments ***

* Exposition *

2) The placement of reference labels is somewhat unpolished with spaces often missing: "on.[22,23]"

=> Thank Reviewer 2 for the kind advice. We corrected unpolished spaces of the manuscript as the advice.

3) The method suddenly introduces concept of voxels without clarifying what this refers to (L43). Typically, voxels would be an element of a discretized volume but no such concept has been suggested here. Furthermore, how does one propagate a voxel (L135)?

=> We apologize for the misleading word. We used the word, voxel, as a floating point in 3D space. We changed the word "propagated voxels" to "*propagated points*" and "voxel" to "*pixel at different depth*".

4) Minor issues:

- L151: "Non-linearity and a wide receptive field are more important than hidden features so the number of channels" - How has this been determined/validated?

=> To compare different models, we varied number of channels and number of hidden layers. The following table summarizes the benchmark results of different models.

Type	PSNR	SSIM	vgg16	alex	GFLOPS
c48	32.3±0.94	0.943±0.010	0.0864±0.0072	0.0523±0.0074	396.8
c24	33.3±0.24	0.955±0.0018	0.0790±0.0021	0.0436±0.0017	187.9
c12	33.1±0.10	0.954±0.00072	0.0811±0.00066	0.0458±0.00062	91.5
c6	32.8±0.11	0.950±0.00088	0.0850±0.00094	0.0494±0.00059	46.6

c48 refers 48 channels with 10 hidden layers, c24 refers 24 channels with 18 hidden layers, c12 refers 12 channels with 34 hidden layers, and c6 refers 6 channels with 68 hidden layers. FLOPS refers floating-point operations per second. The metrics are evaluated on the FHD resolution dataset. Among 4 trained models for each type, one with the worst PSNR is excluded and 3 models are used in averaging. The mean metrics of c12 is different from the metrics of DEHNet because we selected the best model among c12 models. Models, c24 and c12, show better image quality than other models. Since 6 is the minimum number of channels to express complex 3-color field, numbers of channels 24 and 12 is not much larger than the minimum value. In this perspective, we wrote the sentence. We added above table in the supplementary material.

- L198: "Although it was not mentioned" - why not specify it directly in the exposition then?

=> We want to concentrate on explaining synthesis method of DEH at the front part of the manuscript. The intention of the sentence (L198) was to emphasize the presented figures (Fig2 and Fig 3) can be synthesized in real time. We were worried about someone may think that the result gets much worse after quantization.

As suggested by Reviewer 2, we mentioned quantization while introducing DEHNet by using the following sentence.

After training, the weights of the network were quantized to 8-bit integers to reduce computational load.

- The PSNR values should include units (dB).

=> Thank Reviewer 2 for the advice. We added dB to the end of PSNR values as the advice.

* Related work *

5) The the extent of my knowledge the authors discuss relevant work. Additionally, the authors should cite another recent method for high-quality 3D holography:

Lee, Byoung-hyo, et al. "High-contrast, speckle-free, true 3D holography via binary CGH optimization." Scientific reports 12.1 (2022): 1-12.

This work also considers focal stack when optimizing the target hologram. The authors should discuss how different their approach is.

=> The paper [Scientific reports 12.1 (2022): 1-12] used a multiple number of wave field to realize focal stack, so called time-multiplexing method, where DEH only requires one wave field to reconstruct focal stack. The most significant side effect of time-multiplexing method is its hardness of real-time synthesis. To reconstruct scene of focal stack composed of 24 different binary holograms in real time, the conversion rate of each binary hologram should be as fast as 24x60 Hz. Moreover, the paper [Scientific reports 12.1 (2022): 1-12] only propose synthesis method using optimization not network-based synthesis, presumably conversion rate of it would be similar to that of optimization-based DEH, which is slower than DEHNet.

As for the minor differences, the paper adopted incoherent propagation of light to synthesize focal scenes from RGBD images, but it may produce incorrect defocus blur at the boundary of occluded surfaces. And due to the properties of binary hologram, requiring stack of multiple holograms to reconstruct target intensity, it is not possible to use in some of applications like optical tweezers.

We included the following discussion in the manuscript.

Recently, several researches reported enhanced accommodation of CGH by utilizing a number of wave field, so called time-multiplexing method [38, 39]. Although the time-multiplexing method presents great image quality and enhanced accommodation effect, real-time reconstruction of such hologram requires more than ten times of computation resources compared to our method and thus it is still challenging to realize real-time reconstruction.

* Reproducibility *

6) The authors have provided detailed description and assessment of their system. What I could not find is exact type of their SLM. I also miss such level of description for the exact architecture of their network and the training curriculum (optimizer, learning rate,...). The authors could also clearly indicate what part of the data will be shared with other researchers (if any).

=> We used liquid crystal on silicon (LCoS), IRIS-U62 from MAY Inc. of which resolution is 3840x2160 and pixel pitch is 3.6 um. We used the LCoS as 1080p mode by putting same pixel value in 2x2 nearest pixels to minimize pixel crosstalk originated from its small pixel pitch [Lazarev, Grigory, et al. "LCOS spatial light modulators: trends and applications." *Optical Imaging and Metrology: Advanced Technologies 1* (2012)]. As a result, the LCoS behaves as a FHD resolution amplitude-only LCoS with a pixel pitch of 7.2 um.

We used Adam optimizer with a learning rate 0.0005. We reset internal parameters of the optimizer for every 50 epochs at the second stage of training (after fusing normalization layer). Batch size was 16 and the weights of network are updated after 4 batch runs, yielding effective batch size 64.

An exact architecture of our network is just repeat of the unit structure, Conv2d - PReLU - Conv2d - PReLU with identity shortcut, except for the first unit. For the first unit, shortcut is Conv2d layer with kernel size 1 and the number of channels in Conv2d layer is changed from the number of input channels (4) to the number of output channels (12). After concatenating the result of the last layer and the input layer, the channel size is reduced to 6 by a Conv2d layer with kernel size 3.

We updated above information in the manuscript and the supplementary material.

Due to our company policy, we cannot share our dataset publicly. However, since we used open source(<https://github.com/DLR-RM/BlenderProc>) to render training/validation dataset, we could share configuration settings used in synthesizing training/validation dataset. We added the following sentence to "Data availability".

The configuration settings of BlenderProc used in synthesizing training and evaluation dataset will be publicly available along with the paper.

* Technical aspects and validation *

7) What is the purpose of the explicit all-in-focus loss component? The objects at focal plane should already be focused in the respective focal stack layers so why the special treatment? It seems to cause additional complications as discussed on L334. I miss any study of the behavior

that this attempts to mitigate (e.g. ablation).

=> If there exists a solution perfectly reconstructing different 21 planes with one wavefield, the term with beta factor would not be required. As Reviewer 2 indicated, we employed beta factor term to balance sharpness of in-focus objects and defocus blur of out-of-focus objects. We chose 20 as a beta factor to apply similar weights on 20 defocused images and a single focused image. Moreover, the value, 20, makes PSNR of in-focus objects and out-of-focus objects similar as expected. We analyzed all-in-focus(AIF) PSNR and varifocal PSNR depending on beta factor (Please see the following Figure). As beta increases, AIF PSNR increases and varifocal PSNR decreases.

Varifocal PSNR is defined as an average PSNR of 21 planes and AIF PSNR is calculated by propagating the hologram for every unique depth of input depth map and comparing the pixels at that depth with an all-in-focus image, where the depth values in input depth map can be 256 different values. Error bars represent standard deviations of models trained with the same conditions. Among 4 trained models for each condition, one with the worst PSNR is excluded and 3 models are used in averaging.

However, AIF PSNR can be inaccurate when the defocus blur comes into the image. For instance, when the focal plane is at the rear plane, a rear object is occluded by defocus blur of front object. However, blurred front object is bigger than the original size of all-in-focus image due to defocus blur. As a result, even if the rear object is in focus, it is more natural to have different values from the all-in-focus image for some part of the rear object at the defocus blur of the front object. Such incorrect values are common when multiple objects are located at different depths and the circumstance is close to our rendered scenes. (The reason is related with Figure Extended 4)

We included the above information in the supplementary material.

8) Unlike in some of the other methods [15,16], the authors do not force the phase field to be smooth. They demonstrate that this does not pose significant issues for their amplitude-only prototype but I wonder if this would lead to increased artifacts for a phase-only SLM where a suitable coding strategy had to be used (e.g. double phase encoding).

=> As Reviewer 2 indicated, we do not apply smooth phase condition on the field. As far as we know, noise of non-smooth phase is induced because an aperture stop blocks high frequency

components. For instance, if the field includes high frequency components, then high frequency components may have similar frequency to that of a grating used in encoding process. In that case, not only does the optical filter block encoding grating, but it also blocks high frequency components. As a result, hologram is partially lost and the loss induces noise on reconstructed intensity.

In an aspect of using a grating and an optical filter, amplitude-only hologram would also suffer defects if the hologram contains high frequency components. In our opinion, synthesized DEH does not include high frequency components and we expect our DEH can be implemented on phase-only SLMs by adopting double phase encoding method.

9) The captured images presented in Figure 4 do not exhibit as excellent defocus quality as the simulated images. It is my understanding that the (almost) PSF in Figure 1f has been simulated. Would it be possible to provide the same PSF comparison captured on a real setup? It is not clear how the theoretical benefit of arbitrarily optimized phase transfers to an amplitude-encoded and physically constrained real display device.

=> As Reviewer 2 mentioned, Figure 1f is simulated image and Figure 3 (Figure 4 would be typo.) shows less blur compared to simulated images. Since the maximum diameter of defocus blur is fixed to 15 pixels for 3-diopter depth in our configuration, high resolution images show less blur than low resolution images if the sizes of the images are same. To emphasize defocus blur, we moved objects to the maximum and minimum depth of the scene and did experiment with it. Moreover, we used center cropped images in Figure 3. (Please see the following figure)

As far as we know, the benefit in actual device is expected to be the same as theoretical benefit. We also did another experiment to check the benefit using the image used in Figure 1f (Please see the following figure). Since conversion from an amplitude-only hologram (or a phase-only hologram) to complex field requires an optical filter and the optical filter cuts off high frequency components, squares with a few pixels (3, 4, and 6) cannot be reconstructed sharply. Except for the sharpness of the squares, the defocus patterns of experimental results seem similar to those of simulation results.

10) How can the method account for light contribution from scene points occluded in the pinhole RGBD input image?

=> In a 3D scene, occluded surface affects the edge of defocus blur of a front object when the focal plane is at the rear side. Light from the occluded surface of a rear object partially participates in the defocus blur of the front object. However, there is no information of the occluded surfaces in the pinhole RGBD image and thus partial participation of occluded object is hard to consider in hologram synthesis.

Since the occluded surface is usually assumed to be empty during the synthesis of nCGH and so the line near the edge is displayed with background color (*i.e.* black) at the defocus blur of the front object. (Please see the following Figure) In contrast, if we use the whole information of the scene by including the complete red circle during the nCGH synthesis, the edge line diminishes. Considering the fact that there is no black line near the edge of the front object in the reconstructed image of DEHNet, it seems like DEHNet utilizes information near the occluded surface and remove such edge line although the detailed mechanism is hard to confirm.

We updated the above information in the supplementary material.

11) The PSNR is most certainly not a "perceptual image quality" metric (L172). The authors could consider adding a perceptually validated metric such as LPIPS.
=> As commented by Reviewer 2, since only structural similarity (SSIM) can be regarded as a "perceptual image quality" metric, we removed the word, "perceptual" from the sentence. Moreover, as Reviewer 2 recommended, we added LPIPS metrics including vgg-16 and alex in the supplementary material (Please see the following table). And we added the following sentence in the manuscript. Lower is better for vgg16 and alex metrics.

Benchmark results with various image quality metrics including learned perceptual image patch similarity metrics can be found in the supplementary material.

Type	PSNR (dB)	SSIM	vgg16	alex	Time (s)
DEH	33.4±0.61	0.955±0.0039	0.0743±0.0048	0.0415±0.0038	137±0.024
DEHNet(fp32)	33.4±0.57	0.957±0.0035	0.0778±0.0047	0.0427±0.0037	0.0609± 2.7·10 ⁻⁵
DEHNet(fp16)	33.4±0.57	0.956±0.0035	0.0778±0.0047	0.0427±0.0037	0.0287± 4.1·10 ⁻⁵
DEHNet	32.7±0.57	0.939±0.0040	0.126±0.0080	0.0542±0.0039	0.0159± 1.8·10 ⁻⁴
nCGH	27.1±0.59	0.889±0.0079	0.166±0.0088	0.136±0.0078	1.11±0.047
Tensorholo(fp16)	29.9±0.65	0.918±0.0063	0.124±0.0062	0.0898±0.0058	0.0403±0.0011
dCGH	8.06±0.59	0.306±0.0350	0.637±0.028	0.911±0.041	1.71±0.0033

12) Which nCGH method has been used in the comparisons? The timings in Figure 4 seems to assume no neural acceleration has been available for the nCGH, yet previous work has demonstrated that such approach is possible and leads to real-time performance [16]. Were the comparisons done using similarly scaled approaches?

=> We used the method of Ref [16] where its encoding method is changed to Burch encoding method (Ref [39]) to implement the hologram on an amplitude-only SLM. Since the nCGH algorithm does not utilize neural acceleration, we additionally did benchmark with Ref [16] as Reviewer 2 suggested. The following table summarizes the results. (It is the same table with the presented table in the previous comment.) "Tensorholo(fp16)" refers the float16 quantized version of Ref [16] where the quantization is processed by TensorRT as the reference.

Type	PSNR (dB)	SSIM	vgg16	alex	Time (s)
DEH	33.4±0.61	0.955±0.0039	0.0743±0.0048	0.0415±0.0038	137±0.024
DEHNet(fp32)	33.4±0.57	0.957±0.0035	0.0778±0.0047	0.0427±0.0037	0.0609± 2.7·10 ⁻⁵
DEHNet(fp16)	33.4±0.57	0.956±0.0035	0.0778±0.0047	0.0427±0.0037	0.0287± 4.1·10 ⁻⁵
DEHNet	32.7±0.57	0.939±0.0040	0.126±0.0080	0.0542±0.0039	0.0159± 1.8·10 ⁻⁴
nCGH	27.1±0.59	0.889±0.0079	0.166±0.0088	0.136±0.0078	1.11±0.047
Tensorholo(fp16)	29.9±0.65	0.918±0.0063	0.124±0.0062	0.0898±0.0058	0.0403±0.0011
dCGH	8.06±0.59	0.306±0.0350	0.637±0.028	0.911±0.041	1.71±0.0033

*Ref [16]: Maimone, A., Georgiou, A. & Kollin, J. S. "Holographic near-eye displays for virtual and augmented reality." *ACM Transactions on Graphics* (Tog) **36**, 1–16 (2017).

*Ref [39]: Burch, J. "A computer algorithm for the synthesis of spatial frequency filters." *Proceedings of the IEEE* **55**, 599–601 (1967).

13) How were the defocus-blurred depth maps obtained (L335)? The Extended Figure 4 suggests that some form of random sampling has been utilized but the application of said depth maps suggest that a soft depth gradient is intended. Have the maps been additionally filtered to that goal?

=> We apologize for misleading words. We expressed the word as “defocus-blurred depth map”, but “defocus-blur-considered depth map” would be more appropriate for this depth map. We want to synthesize a depth map including defocus blur where the depth value at the defocus blur is same with the depth value of a blurred front object. However, if we include defocus blur when rendering depth map, soft depth gradient exists at the defocus blur of the depth map. This is because small portion of light from the rear object occluded by the front object is also recorded at the edge of the blurred front object.

Defocus-blur-considered depth map is used when calculating all-in-focus loss and such averaged depth map would mislead all-in-focus loss as the blurred object exists at an intermediate plane. To prevent those circumstances, we sampled the depth map using only one ray per each pixel and collect 10 depth maps for each focal distance. Among 10 depth values of each pixel, only the front-most depth value is used, so as the depth values of the blurred pixels are confined to the front depth. We changed the word “defocus-blurred depth map” to “defocus-blur-considered depth map” and added the following explanations in the methods.

More specifically, we synthesized a depth map including defocus blur where the depth value at the defocus blur is same with the depth value of a blurred front object.

14) Was the method [1] in the supplement trained on the same data as the proposed method?

=> We used the original network of the method [1]. Hologram synthesis code of the method [1] is not shared and thus converting our image dataset to their holograms was impossible.

Moreover, textures of our training dataset(CC0 textures, <https://cc0textures.com/>) are different from the textures of evaluation dataset([40–43]) so we considered the evaluation dataset is far different from the training dataset. The following figure presents example images and depth maps of different datasets. All-in-focus images and depth maps of training dataset (a, b), 512 resolution evaluation dataset (c, d), and FHD resolution evaluation dataset (e, f) are shown. We updated the example images in the supplementary material.

*Ref [1]: Shi, L., Li, B., Kim, C., Kellnhofer, P. & Matusik, W. “Towards real-time photorealistic 3D holography with deep neural networks.” *Nature* **591**, 234–239 (2021).

Reviewer #3 (Remarks to the Author):

The authors present a method to generate multiplane holograms of 3D color scenes with improved depth perception and non-diffuse phase. In particular, their method solves one of the main issues with non-diffuse phase holograms, namely the lack of natural blur in the defocused regions of the reconstructed scene. To achieve this, they make use of a physically-based renderer that simulates the adequate defocus blur at different depths of the scene and then feeds this render and the color and depth information of the scene to a convolutional neural network, which is trained to generate the adequate field distribution. Once the neural network is trained, only the color and depth map of a scene is required to generate its hologram. The presented results are remarkable, dealing with a relatively unexplored issue, and the paper is well written. In particular, adequate depth blur is reproduced at different depths after reconstructing the generated hologram, compared to the lack of such blur in common in conventional multiplane hologram generation. This has the potential to lead to improvements in holographic displays and associated applications.

Nevertheless, some important points should be addressed.

=> Before we respond to Reviewer 3's questions and comments, we would like to thank Reviewer 3 for making thoughtful comments and important questions. In particular, we would like to thank Reviewer 3 for bringing up an important issue, maximum scene depth of DEH. We used 3 diopter as maximum scene depth since we thought hand interaction distance of AR/VR applications is around 3 diopter. Thanks to Reviewer 3, we can clarify the upper bound of the maximum scene depth of DEH.

1. The results in this paper use 21 varifocal images to generate the holograms, however, using as a criterion that "the number of varifocal images was selected to be 21 pixels larger than the maximum diameter of the blur circle". I believe this point is essential to understanding the performance of the method, and as such the authors should expand the associated discussion. Some relevant questions are

- How many individual planes must be rendered to achieve a hologram with optimal depth blur?
- Does the number of planes depend on the target depth of the scene?

=> Defocus blur is induced by propagation of incoherent light and thus the diameter of defocus blur is proportional to the distance between the object and the focal plane. If a distance between two nearest individual planes corresponds to the distance producing one-pixel-diameter of defocus blur, then the intensity shift by propagation of incoherent light between the two nearest individual planes would be 1 pixel. In order to set enough target images for pixel-level accuracy, we set the number of individual planes to the number which is larger than the maximum diameter of defocus blur of the scene.

In this sense, the number of individual planes depends on the maximum defocus blur derived from the maximum depth of scene we want to reconstruct. In the manuscript, maximum diameter of blur circle is 15 pixels when the maximum depth of presented scenes is 3 diopter (Related information can be found in "Determining the diameter of the blur circle" section of Methods). To ensure pixel-level accuracy, the number of planes are selected as 21, larger than the maximum diameter of the blur circle. In the same way, if the maximum depth is selected as 5 diopter, then at least 25 planes are required to ensure pixel-level accuracy.

We added above information in a new section of the supplementary material, "Requirements for the number of target planes".

-Is the number of individual planes fixed after training the neural network?

=> Yes, it is fixed after the training. However, the number of planes is sampled enough and image difference between nearest individual planes is smaller than 1 pixel. As a result, the network accepts 0~255 depth values and synthesizes appropriate holograms for every depth in pixel-level accuracy.

- Which is the maximum scene depth that can be successfully reconstructed with the proposed method?

=> Maximum scene depth under the current parameters is 3 diopter. However, the maximum scene depth can be arbitrarily increased. The side effect of increasing the maximum scene depth is decreased image quality since the number of individual planes that hologram should reconstruct increases. We set the number of planes per diopter to 6.67 planes/diopter as in the manuscript and analyzed image quality depending on the maximum scene depth using optimization method to avoid the effect coming from capability of a neural network.

To evaluate image quality metrics depending on the maximum scene depth, we synthesized 200 scenes with maximum scene depth 9 diopters and 61 focal planes. The 61 plane dataset was synthesized using the same method as the previous dataset and its resolution was FHD. When optimizing and evaluating the hologram with smaller maximum scene depth than 9 diopter, subset of 61 planes are used. For instance, 31 planes are used for optimizing and evaluating DEH with 4.5 diopter, and 41 planes are used for 6 diopter DEH. By using the same scenes in evaluating the DEHs possessing different maximum depths, it is possible to evaluate relative difference between them.

The following figure presents image quality metrics depending on the maximum scene depth. As maximum scene depth increases image quality metrics (PSNR and SSIM) decrease. Error bar indicates standard deviation between scenes. If we set the image quality criteria as PSNR>30 dB, then 4.5 diopter is the maximum scene depth we can use. However, the criteria can be arbitrarily selected and someone may use DEH with 6.0 diopter if the image quality is not the top priority.

We added such information in a new section of the supplementary material, "Maximum scene depth of DEH".

- In conventional multiplane holograms, the computation time has a strong relationship with the number of individual planes. Is this the case with this approach?

=> As other network-based methods, computation time does not depend on the number of individual planes after training and 255 depths can be processed if the number of planes of training dataset is properly decided.

2. Did the authors measure if there is any difference in the diffraction efficiency of the holograms generated with gradient descent and the neural network? This is particularly important since the use of amplitude-only SLM already implies a lower diffraction efficiency compared to phase-only modulation.

=> Efficiency of diffraction-engineered hologram is slightly higher than the conventional amplitude-only holograms. For FHD resolution dataset, experimentally measured efficiency of DEH and DEHNet is 9.5% higher than the measured efficiency of nCGH. The following table summarizes experimentally measured relative efficiencies. Here, the measured efficiencies are normalized to the average efficiency of nCGH and image difference in the dataset causes standard deviation of efficiency.

Type	nCGH	DEH	DEHNet
Normalized efficiency	1	1.095	1.095
Standard deviation	0.099	0.106	0.104

However, amplitude-only holograms have much lower efficiency than phase-only holograms as Reviewer 3 commented and efficiency of amplitude-only holograms should be further increased in a long run.

3. The introduction lacks an overview of the recent work in multiplane hologram generation. Remarkably, there is only a single recent citation regarding this point [26], and most are from 2010 and before. The authors should consider a careful review of the recent literature to update the bibliography.

Some relevant references

- [1] C. Chen, B. Lee, N.-N. Li, M. Chae, D. Wang, Q.-H. Wang, and B. Lee, "Multi-depth hologram generation using stochastic gradient descent algorithm with complex loss function," *Opt. Express* 29, 15089–15103 (2021)
- [2] J. Zhang, N. Pégard, J. Zhong, H. Adesnik, and L. Waller, "3D computer-generated holography by non-convex optimization," *Optica* 4, 1306–1313 (2017)
- [3] P. Zhou, Y. Li, S. Liu, and Y. Su, "Dynamic compensatory Gerchberg–Saxton algorithm for multiple-plane reconstruction in holographic displays," *Opt. Express* 27, 8958–67 (2019)
- [4] A. Velez-Zea, J. Fredy Barrera-Ramírez, and R. Torroba, "Improved phase hologram generation of multiple 3D objects," *Appl. Opt.* 61, 3230 (2022)
- [5] C. Ying, P. Hui, C. Fan, and W. Zhou, "New method for the design of a phase-only computer hologram for multiplane reconstruction," *Opt. Eng.* 50, 055802 (2011)

=> We agree with the suggestion of Reviewer 3. We introduced some of recent progress in multiplane hologram as Reviewer 3 suggested. The following sentences are newly introduced sentences in the manuscript.

Multi-plane hologram attracts great attention recently, especially for its improvement on image quality and computation time [1-5]. For instance, non-convex optimization is adopted to minimize a custom cost function [2], dynamic adjustment of amplitude-constraint is employed to improve image quality [3], and a new algorithm based on singular value decomposition of the Fresnel impulse response function is proposed to enhance computational speed [5].

4. The loss function used in this work is a per-pixel error metric, however, this is not necessarily the best approach when improved perception of the hologram reconstruction is desired. An alternative better suited for human perception is the structural similarity index (SSIM). Did the authors test or consider alternative loss functions? Some discussion regarding this point would

clarify this point. A relevant reference is Yang, F., Kadis, A., Mouthaan, R. et al. Perceptually motivated loss functions for computer-generated holographic displays. Sci Rep 12, 7709 (2022). <https://doi.org/10.1038/s41598-022-11373-8>

=> We tried to use perceptual loss, but its effect was unclear. We did experiments with the DEH optimized by multi-scale structural similarity (MS-SSIM) loss, which gives the best results in [Sci Rep 12, 7709 (2022)]. The following figure presents the results and DEH-MS in the figure refers the reconstructed image of the DEH optimized using MS-SSIM loss.

Although MS-SSIM loss gives the best image quality in single plane holograms, DEH optimized by MS-SSIM loss suffers from some defects. For instance, defocus blur seems inappropriate at the letters of a can in the MS-SSIM result (enlarged image of rear focus in the following figure).

In contrast to single plane holograms, which were investigated in the research, further investigation of loss functions could be another research topic for multi-plane holograms.

We included the above information in the supplementary material.

5. Figure 1 is extremely important in the present manuscript, showing how the different methods deal with the phase information, the rendering process, the DEH method, the convolutional neural network, and the in-focus and out-of-focus intensities in each method. Both the figure and caption are extremely dense, and the font size in the figure is very small, making its reading and interpretation difficult. I believe that each of these points merits an individual figure with their corresponding discussion and caption, or at least a larger size for each subfigure.

=> We agreed with Reviewer 3's comment, so we divide Figure 1 to figures as Reviewer 3 suggested. The following 2 figures are divided figures.

6. The labels in figure 2 are difficult to read due to the poor contrast with the object. This should be addressed, for example by using a black outline for the labels. Consider also using letter labels to refer to each figure instead of their position (bottom-left, top-left). This would make the caption easier to read.

=> We thank Reviewer 3 for the suggestions. We changed the labels and the captions as Reviewer 3 suggested.

7. The indents of the zoomed-in regions showing the depth effects in Figures 2, 3, and the extended figures 1 and 2 should be larger to allow a better perception of the difference between the methods. This should also be done in the figures in the supplementary material, in particular, figure S5 lacks zoomed indents that allow perceiving the effect.

=> We increased the total size of Figure 2,3 for 20%. Especially for Figure 3, we moved objects to the maximum and minimum depth of the scene and did experiment with it. Moreover, we replaced the images with center cropped images and increased magnification of the zoomed indents of Figure 3. We also increased the magnification of zoomed-in regions of Figures 2, 3,

and the extended figures 1 and 2. Additionally, we increased the size of figures in supplementary material and added zoomed indents of Figure S5.

8. The authors should provide the model of the SLM used in the experiment.

=> We used liquid crystal on silicon (LCoS), IRIS-U62 from MAY Inc. of which resolution is 3840x2160 and pixel pitch is 3.6 μm . We used the LCoS as 1080p mode by putting same pixel value in 2x2 nearest pixels to minimize pixel crosstalk originated from its small pixel pitch [Lazarev, Grigory, et al. "LCOS spatial light modulators: trends and applications." *Optical Imaging and Metrology: Advanced Technologies 1* (2012)]. As a result, the LCoS behaves as a FHD resolution amplitude-only LCoS with a pixel pitch of 7.2 μm . We updated above information in the Method.

9. To better show the effect of the depth blur, the authors should consider including a video of both the numerical and experimental reconstruction of one of the scenes as the focus plane is changed.

=> We produced animations of numerical and experimental reconstruction as Reviewer 3 suggested.

Given the above points, I believe this paper can only be considered for publication after major revisions.

REVIEWER COMMENTS

Reviewer #1 (Remarks to the Author):

All my questions and comments have been considered properly. The work is original and the methodology is sound. But the highlights of the proposed method should be fully verified and demonstrated by a desired optical experiments.

The Figure Extended 3 is of low quality for the NC. It is strange to use lens group in the figure and use lens array in the caption.

The reconstruction results in Figure Extended 4 are not impressive because the reconstruction resolution is very low and there are only two layers. It is strongly suggested to improve the experimental results to improve the impact of this work. One solution is to use the high resolution phase only SLM. The quality and the resolution are pretty limited when using an amplitude only SLM.

The whole work should be reconsidered to compare with a newly published works by MIT.

Liang Shi, Beichen Li, and Wojciech Matusik, "End-to-end learning of 3D phase-only holograms for holographic display," *Light: Science & Applications* 11, 247 (2022).

Reviewer #2 (Remarks to the Author):

The authors have made a clear effort to address the comments of me and the other two reviewers. I do not observe any remaining serious concerns in my review or the other reviews. Therefore, I maintain my positive view of the submission and recommend accepting the paper.

Reviewer #3 (Remarks to the Author):

The authors have addressed satisfactorily all the points raised in the first revision of the manuscript. There are, however, some minor points that should be addressed.

-Some of the changes to the text arising from the revision of the manuscript could benefit from a careful review. For example, in line 84-86, the phrase “As the multi-plane hologram is synthesized by optimizing wavefield to resemble one image at one distance while resemble other images at other distances, wavefield of our hologram is optimized to reconstruct different blurred images and sharply focused images depending on a propagation distance” is difficult to understand. Another example can be found in lines 352-354 “More specifically, we synthesized a depth map including defocus blur where the depth value at the defocus blur is same with the depth value of a blurred front object” is also very confusing.

-The authors should consider directly referencing and discussing in the manuscript some of the results added in the supplementary material, including the videos of the reconstruction experiment.

Despite the above points, I believe that this paper is now suitable for publication.

Reviewer #1 (Remarks to the Author):

All my questions and comments have been considered properly. The work is original and the methodology is sound. But the highlights of the proposed method should be fully verified and demonstrated by a desired optical experiments.

=> We appreciate Reviewer 1 for helpful comments. We can increase quality of the manuscript thanks to the helpful advice of Reviewer 1.

The Figure Extended 3 is of low quality for the NC. It is strange to use lens group in the figure and use lens array in the caption.

=> We appreciate Reviewer 1 for indicating the typo and the suggestion. We replaced the Figure extended 3 with the following figure as Reviewer 1's suggestion. And the words "lens array" in the caption are replaced with the words "lens group".

The reconstruction results in Figure Extended 4 are not impressive because the reconstruction resolution is very low and there are only two layers. It is strongly suggested to improve the experimental results to improve the impact of this work. One solution is to use the high resolution phase only SLM. The quality and the resolution are pretty limited when using an amplitude only SLM.

=> We apologize for the misleading words. In the Figure extended 4, we tried to explain why the defocus-blur-considered depth map is required in our dataset. For the explanation, we enlarged a rendered image to clearly present the boundaries of all-in-focus depth map and defocus-blur-considered depth map. As a result, only 140x140 resolution image is presented in the figure. To prevent misinterpretation, we marked "All-in-focus rendered image. White box indicates enlarged area for other subfigures." in the caption and added uncropped image in the Figure extended 4 (please see the following figure).

The whole work should be reconsidered to compare with a newly published works by MIT.

Liang Shi, Beichen Li, and Wojciech Matusik, "End-to-end learning of 3D phase-only holograms for holographic display," *Light: Science & Applications* 11, 247 (2022).

=> We appreciate Reviewer 1 for kind advice. The research [*Light: Science & Applications* 11, 247 (2022)] solved occlusion artifact problem using dataset composed of layered depth images (LDIs). By efficiently representing 3D scenes using LDIs, they avoided lack of information of occluded objects. Such lack of information is also already described in Sec. 10 of our supplementary material.

Although the occlusion problem is solved in Ref [*Light: Science & Applications* 11, 247 (2022)], the authors didn't deal with amount of defocus blur, one of our main purposes in our manuscript.

To introduce the Ref [*Light: Science & Applications* 11, 247 (2022)], we added the following sentences in *Discussions* section of the manuscript.

"Besides, another approach is recently proposed to remove occlusion artifacts by adopting a layered depth image in learning-based CGH algorithms. However, the approach did not deal with amount of defocus blur, distinctive from our method."

Reviewer #2 (Remarks to the Author):

The authors have made a clear effort to address the comments of me and the other two reviewers. I do not observe any remaining serious concerns in my review or the other reviews. Therefore, I maintain my positive view of the submission and recommend accepting the paper. => We thank Reviewer 2 for recommending our manuscript and helpful comments in the last revision.

Reviewer #3 (Remarks to the Author):

The authors have addressed satisfactorily all the points raised in the first revision of the manuscript. There are, however, some minor points that should be addressed.

=> We appreciate Reviewer 3 for kind advice. We can rewrite confusing sentences thanks to the helpful comments of Reviewer 3.

-Some of the changes to the text arising from the revision of the manuscript could benefit from a careful review. For example, in line 84-86, the phrase "As the multi-plane hologram is synthesized by optimizing wavefield to resemble one image at one distance while resemble other images at other distances, wavefield of our hologram is optimized to reconstruct different blurred images and sharply focused images depending on a propagation distance" is difficult to understand. Another example can be found in lines 352-354 "More specifically, we synthesized a depth map including defocus blur where the depth value at the defocus blur is same with the depth value of a blurred front object" is also very confusing.

=> We appreciate Reviewer 3 for careful review. We replaced the sentences with the following sentences.

In line 84~86,

"As the multi-plane hologram is synthesized by optimizing wavefield to resemble one image at one distance while resemble other images at other distances, wavefield of our hologram is optimized to reconstruct different blurred images and sharply focused images depending on a propagation distance"

=> *"The multi-plane hologram is synthesized by optimizing wavefield to reconstruct one image at one focal plane while optimized to reconstruct other images at other focal planes. In the same way, the wavefield of our hologram is optimized to reconstruct sharply focused image of an object at the object plane and reconstruct blurred images at other focal planes."*

In line 359-363,

"More specifically, we synthesized a depth map including defocus blur where the depth value at the defocus blur is same with the depth value of a blurred front object"

=> *Normally, if we include defocus blur when rendering depth map, front depth values and rear depth values are blended at an edge of defocus blur of a front object. Instead, we sampled the depth map using only one ray per each pixel and collected 10 depth maps for each focal distance. Among 10 depth values of each pixel, only the front-most depth value is used, so as the depth values of the blurred pixels are confined to the depth of the front object.*

-The authors should consider directly referencing and discussing in the manuscript some of the results added in the supplementary material, including the videos of the reconstruction experiment.

=> We marked specific sections of the supplementary material when referencing the supplementary material in the manuscript as Reviewer 3 suggested. More specifically, we added numbers to the section titles of the supplementary material and marked the number when referencing the supplementary material, e.g. "see Sec. 9 of the supplementary material for further details of all-in-focus loss". And we introduced the supplementary videos in the manuscript with the sentence, "Numerical and experimental reconstruction of nCGH and DEH with changing focal plane can be found in Supplementary videos.". Moreover, we newly referred some sections of the supplementary material in the manuscript and added details while referencing the supplementary material.

Despite the above points, I believe that this paper is now suitable for publication.